# Neural ODE Transformers: Analyzing Internal Dynamics and Adaptive Fine-tuning

**Anh Tong**
Korea University
anhtong12@korea.ac.kr

**Thanh Nguyen-Tang**
Johns Hopkins University
nguyent@cs.jhu.edu

**Dongeun Lee**
Texas A&M University-Commerce
dongeun.lee@tamuc.edu

**Duc Nguyen**
Qualcomm AI Research*
ducanguy@qti.qualcomm.com

**Toan Tran**
Qualcomm AI Research*
ttran@qti.qualcomm.com

**David Hall**
Stanford
dlwh@stanford.edu

**Cheongwoong Kang**
KAIST
cw.kang@kaist.ac.kr

**Jaesik Choi**
KAIST, INEEJI
jaesik.choi@kaist.ac.kr

## Abstract

Recent advancements in large language models (LLMs) based on transformer architectures have sparked significant interest in understanding their inner workings. In this paper, we introduce a novel approach to modeling transformer architectures using highly flexible non-autonomous neural ordinary differential equations (ODEs). Our proposed model parameterizes all weights of attention and feedforward blocks through neural networks, expressing these weights as functions of a continuous layer index. Through spectral analysis of the model's dynamics, we uncover an increase in eigenvalue magnitude that challenges the weight-sharing assumption prevalent in existing theoretical studies. We also leverage the Lyapunov exponent to examine token-level sensitivity, enhancing model interpretability. Our neural ODE transformer demonstrates performance comparable to or better than vanilla transformers across various configurations and datasets, while offering flexible fine-tuning capabilities that can adapt to different architectural constraints.

## 1 Introduction

The recent advancements in large language models (LLMs) based on transformer architectures (Vaswani et al., 2017; Brown et al., 2020) have impressive empirical performance in various tasks. These transformer models with millions to billions of parameters, pose a formidable challenge in comprehending the intricate processes transpiring within their layers. Initial efforts have been undertaken to unravel mechanistic interpretability (Elhage et al., 2021) and adopt a mathematical approach (Geshkovski et al., 2023a;b) to better understand the inner workings of these complex models.

As the field of LLMs continues to evolve and grow, it is important to develop a fundamental approach to understanding such complex models. This can provide stepping stones for improving trustworthiness, interpretability, and alignment in LLMs (Shevlane et al., 2023). To this aim, there are multifaceted approaches combining insights from mechanistic interpretability, mathematical analysis, and other complementary methods. Rigorous approaches from Geshkovski et al. (2023a;b) open

---

*Qualcomm Vietnam Company Limited

interesting directions, providing theoretical understanding of transformers using neural ordinary differential equations (neural ODEs) (Chen et al., 2018). In practical applications, neural ODEs have been used in models proposed by Lu et al. (2019); Zhong et al. (2022); Dutta et al. (2021). While Geshkovski et al. (2023a;b) suggest that the model outputs collapse to a small number of points at infinite depth, Zhong et al. (2022); Dutta et al. (2021) demonstrate the use cases of neural ODE-based transformers. However, the main limitation in this line of work is that they often rely on shared-weight assumptions, which leads to a discrepancy between theoretical results and practical observations. Theoretically, the clustering behavior of fixed points in these studies (Geshkovski et al., 2023a;b) does not align well with the outputs of autoregressive transformers, which flexibly predict next tokens rather than forming clusters. Practically, the potential of differential equation-like adaptive computation is often overlooked, as evidenced by the limited exploration of neural ODE versions of transformers with time-dependent vector fields, as opposed to weight-sharing approaches.

In this paper, we introduce a novel approach to modeling transformer architectures using highly flexible non-autonomous neural ODEs[1]. Apart from existing methodologies that seek weight-sharing, our proposed model generates all the weights of attention and feed-forward blocks through hyper-neural networks. Specifically, we consider weights as functions of a continuous layer index (or time). We conduct a comprehensive exploration and analysis of its dynamics through the lens of dynamical systems. Our primary analytical tool is spectral analysis that examines the components of our models including attention blocks and feed-forward blocks. Our analysis reveals some important properties of our models. The observed spectral dynamics tend to exhibit increasing magnitudes of eigenvalues. Through simulation, we demonstrate that this phenomenon inhibits the emergence of clusters, contrasting with the results reported in Geshkovski et al. (2023a). In other aspects, we use the Lyapunov exponent to study token-level sensitivity, which can be a potential tool for explainable machine learning methods (Gunning et al., 2019).

Furthermore, we present a novel approach that leverages the adaptive computation of differential equations to create a versatile pretrained model. Our model exhibits a unique characteristic: the ability to be fine-tuned under various architectures derived from the initial pretrained structure. When presented with novel datasets, our model dynamically adjusts the step size of its ODE solvers. This adaptation effectively transform the model's architecture, allowing it to function as a vanilla transformer. Consequently, this enables the application of a wide range of fine-tuning techniques such as LoRA (Hu et al., 2021). To the best of our knowledge, this represents the first instance of a pretrained model capable of being fine-tuned across diverse architectural configurations.

The contributions of this paper are threefold. First, we introduce neural ODE transformers having comparable performance with GPT in different settings and data sets. Second, we offer a comprehensive analysis of the proposed models, unraveling their intricate internal dynamics and a technique to assess token-level sensitivity, enhancing our ability to interpret model behaviors. Lastly, we show that our model provides flexible computation fine-tune while maintaining a competitive performance.

## 2 REVISITING TRANSFORMER

Transformers proposed by Vaswani et al. (2017) consist of key components, including self-attention and feed-forward neural network.

**Attention mechanism** Consider a sequence with $n$ tokens, $X_1, \ldots, X_n$ in $\mathbb{R}^d$, and represent it as a matrix $X \in \mathbb{R}^{n \times d}$. Let us define three linear projections $Q, K, V \in \mathbb{R}^{d_{\text{attn}} \times d}$ for query, key, and value, respectively. The self-attention operation is expressed as follows:

$$\text{Attn}(Q, K, V; X) = \text{softmax}\left(\frac{XQ^\top KX^\top}{\sqrt{d}}\right)(XV^\top). \tag{1}$$

In essence, the self-attention mechanism acts as an operator determining where to focus attention within the entire sequence. The softmax function introduces emphasis, making it more likely that one token will be highlighted over others.

---

[1]Non-autonomous ODEs have time-dependent vector fields.

Self-attention is further extended into multi-head attention. In this configuration, multiple independent self-attention outputs, referred to as *heads*, are concatenated and reprojected, allowing for the introduction of residual connections. The formal definition is given by:

$$\text{MultiHead} := \text{Concat}(\text{head}_1, \dots, \text{head}_H)O,$$
$$\text{where} \quad \text{head}_i := \text{Attn}(Q_i, K_i, V_i; X), \ i = 1, \dots, H.$$

Here, $H$ is the number of heads and $O \in \mathbb{R}^{d_{\text{concat}} \times d}$ is a projection matrix.

**Feed-forward block**   The second crucial component is the feed-forward layer, modeled as a two-layer multilayer perceptron with a nonlinear activation function, expressed as:

$$\text{GeLU}(XW_1^\top + b_1)W_2^\top + b_2,$$

where $W_1 \in \mathbb{R}^{d_{\text{mlp}} \times d}$, $W_2 \in \mathbb{R}^{d \times d_{\text{mlp}}}$, $b_1 \in \mathbb{R}^{d_{\text{mlp}}}$, and $b_2 \in \mathbb{R}^d$. The feed-forward layer, a singular component in the transformer, introduces nonlinearity in the computational flow, enhancing the model ability to capture complex patterns and relationships.

## 3   RELATED WORK

The work in Lu et al. (2019) is pioneering in establishing the link between neural ODEs and transformers. They recognize that the residual connections in transformer blocks can be interpreted as a Lie-Trotter splitting scheme akin to ODE solvers. Building on this insight, Lu et al. (2019) proposes a novel architecture called MacaronNet, motivated by an enhanced numerical solver. In a parallel vein, Li et al. (2021) presents a new variant of transformers inspired by higher-order solvers to address machine translation tasks. Our approach differs from these works in a key aspect: while they were inspired by ODE solvers to develop novel transformer construction methods, we focus on directly formulating differential equations. Consequently, our model enjoys all the properties of neural ODEs, such as adaptive computation, and is compatible with any ODE solver.

The approach outlined in Dutta et al. (2021) aligns with certain aspects of our proposed model. Specifically, this approach involves introducing time-evolving attention layers and feed-forward layers. In the context of attention mechanisms, Dutta et al. (2021) suggests concatenating inputs and layer embeddings. However, it is worth noting that this approach may need more flexibility, as there persists a shared component in the computation of attention weights. This shared component relies solely on input and remains independent of layer embeddings.

In another investigation, Zhong et al. (2022) explores various configurations of weight-sharing strategies when training transformers in a neural ODE style. The key finding indicates that as the degree of weight-sharing between layers increases, the model's performance tends to degrade. Our model overcomes this limitation by using time-dependent weights, resulting in improved performance.

A body of research has emerged, focusing on constructing models and laying down theoretical frameworks for transformers built upon neural ODEs. In works such as Geshkovski et al. (2023a;b), transformers are conceptualized as particle systems, and the geometric behaviors of simplified equivalents based on neural ODEs are analyzed. The predominant findings in these studies highlight the tendency of systems to converge to clusters as they approach their limits. Interestingly, Sinkformer (Sander et al., 2022) also uses a similar theoretical approach but aims to design a new variant of transformers. However, we note that the weight-sharing assumption of Geshkovski et al. (2023a;b) is rather restrictive. Our work demonstrates that learned weight dynamics prevent models from suffering clustering effects. This suggests to reconsider alternative assumptions for such studies.

Research has been devoted to deep equilibrium models, as demonstrated by Bai et al. (2019), showcasing a close relationship with neural ODEs. This technique has been effectively employed in training transformer-like models with unspecified depth during training. While similar in some respects, our model (and neural ODEs in general) allows for adaptive fine-tuning with flexible ODE solver step sizes.

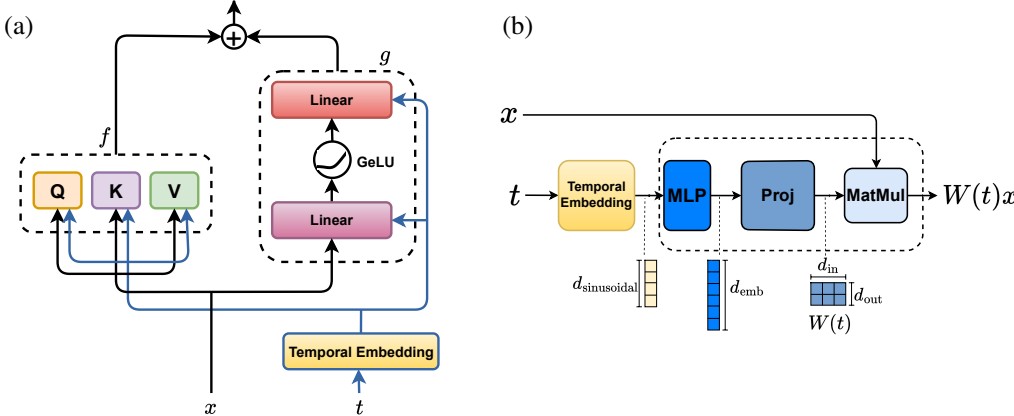

Figure 1: (a) The vector field of DIFFEQFORMER with attention block and feed-forward block constructed from time-dependent weights. (b) The architecture of time-dependent weights.

## 4 TIME-DEPENDENT WEIGHT TRANSFORMER

This section introduces our proposed transformers as differential equations, called DIFFEQFORMER. In the beginning, we formulate the transformer within the context of ODEs. We then provide the approach for representing time-dependent weights in our models.

### 4.1 TRANSFORMERS THROUGH THE LENS OF DIFFERENTIAL EQUATIONS

Consider a sequence $X_1, \ldots, X_n$ with $X_i \in \mathbb{R}^d$. Let a dynamical system evolve the given sequence under the following differential equation:

$$\dot{x}_i(t) = f(x_i(t), x_{[n]}(t), t) + g(x_i(t), t), \qquad t \in [0, T], \qquad (2)$$
$$x_i(0) = X_i, \qquad i = 1, \ldots, n. \qquad (3)$$

Here, $\dot{x}_i(t) = \dfrac{dx_i}{dt}$ and $x_{[n]}(t)$ denotes $[x_1(t), \ldots, x_n(t)]$. This formulation was first presented in Lu et al. (2019). Based on a numerical solver, they proposed the MacaronNet architecture with *discrete* layers, in contrast to our approach which uses *continuous* layers with *time-dependent* weights (see Section 4.2 on time-dependent weights).

Here $f$ acts as an attention mechanism, capturing inter-particle interactions. Self-attention facilitates the transfer of information among particles. In a multi-head setting, each particle receives information from $H$, which is the number of different versions of all particles, and updates itself based on this information. To express this, we define $f$ as follows:

$$f(x_i(t), x_{[n]}(t), t) = \sum_{h=1}^{H} \frac{1}{L_{i,h}} \sum_{j=1}^{n} \exp\left(\frac{\langle Q_h(t) x_i(t), K_h(t) x_j(t) \rangle}{\sqrt{d}}\right) V_h(t) x_j(t), \qquad (4)$$

where $Q_h(\cdot), K_h(\cdot), V_h(\cdot)$ are the mapping from $[0, T]$ to $\mathbb{R}^{d_{\text{attn}} \times d}$. $L_{i,h} = \sum_{j=1}^{n} \exp\left(\frac{\langle Q_h(t) x_i(t), K_h(t) x_j(t) \rangle}{\sqrt{d}}\right)$ is the normalizing term in the softmax function and $\langle x, y \rangle = x^\top y$.

On the other hand, the convection $g$ serves a role as feed-forward blocks having all time-dependent weights:

$$g(x_i(t), t) = W_2^{\text{FF}}(t) \text{GeLU}(W_1^{\text{FF}}(t) x_i(t) + b_1(t)) + b_2(t). \qquad (5)$$

Figure 1a illustrates how to construct the vector field of DIFFEQFORMER.

**Remarks** The particle system exhibits certain similarities with consensus-based optimization techniques, as explored in previous works (Pinnau et al., 2017; Carrillo et al., 2021), with an initial

discussion provided in Geshkovski et al. (2023b). Drawing upon this resemblance, it becomes possible to elucidate specific observations in transformers, revealing that these models implicitly engage in optimization processes (mesa-optimization) over contextual information (Garg et al., 2022; Von Oswald et al., 2023; von Oswald et al., 2023).

Notably, the work of Elhage et al. (2021) introduces the concept of *residual streams*: the connections between input tokens and their corresponding output tokens. In this framework, the role of attention is to facilitate the communication and information transfer between these residual streams. Multi-head attention encompasses information from various subspaces, allowing for the merging and writing of information to residual streams through residual connections. This perspective aligns with the interpretation of particle systems described in Lu et al. (2019); Geshkovski et al. (2023a;b).

Additionally, placing attention layers and feed-forward layers at the same level was investigated in prior studies (Peng et al., 2022; Zhong et al., 2022; Dehghani et al., 2023). This approach allows us to fully conceptualize transformers as neural ODEs with Euler methods, in contrast to the Lie-Trotter scheme employed in Lu et al. (2019).

## 4.2 REPRESENTATION OF TIME-DEPENDENT WEIGHTS

We model time-dependent weights for attention components $Q_h(t)$, $K_h(t)$, $V_h(t)$, and feed-forward weights $W_1^{\text{FF}}(t)$, $W_2^{\text{FF}}(t)$, using a time-dependent unit $W : \mathbb{R}_+ \to \mathbb{R}^{d_{\text{in}} \times d_{\text{out}}}$ that embeds the time information in the Fourier domain, inspired by the existing diffusion model architectures (Song et al., 2020; Peebles & Xie, 2023). In particular, for any time $t \in \mathbb{R}_+$,

$$W(t) = \text{Proj}(\text{MLP}(\text{Sinusoidal}(t))), \tag{6}$$

where Sinusoidal : $\mathbb{R} \to \mathbb{R}^{d_{\text{sinusoidal}}}$ is used to embed time $t$ into a higher-dimensional space and the two-layer multilayer perceptron MLP : $\mathbb{R}^{d_{\text{sinusoidal}}} \to \mathbb{R}^{d_{\text{emb}}}$ is used. The MLP's output will then be linearly projected and reshaped to match the desire of the weight matrices by using the operator Proj : $\mathbb{R}^{d_{\text{emb}}} \to \mathbb{R}^{d_{\text{in}} \times d_{\text{out}}}$. Figure 1b depicts the architecture of $W(t)$.

**Remark** The time-dependent weights can be understood as hypernetworks (Ha et al., 2016; Stanley et al., 2009). However, our approach stems from a distinct motivation, aiming to establish a flexible method for modeling time-dependent vector fields in differential equations, employing a Fourier time embedding representation rather than the architecture proposed in Ha et al. (2016). The framework presented by Ha et al. (2016) contains two types of hypernetworks: static and dynamic. Our approach aligns more closely with the static variant. Conversely, existing work, such as Zhang et al. (2019); Choromanski et al. (2020), in the realm of neural ODE, attempts to parameterize weights using differential equations, resembling a form of dynamic hypernetworks. While this paper does not explore this approach, it is noteworthy that the main challenge lies in considering the state of the original ODE, which is high-dimensional and could result in computational expense.

## 5 ANALYZING INTERNAL DYNAMICS OF TRANSFORMERS

In this section, we explore the qualitative aspects and properties of our proposed model, including its behaviors of attention mechanism and model interpretability. Our analysis presented here is conducted on the DIFFEQFORMER model trained on the OPENWEBTEXT dataset (refer to Section 6 for the detailed setup).

### 5.1 ANALYSIS OF ATTENTION MECHANISM

To characterize attention blocks in DIFFEQFORMER, we will use existing tools, including gradient flows (Geshkovski et al., 2023a;b) and induction heads (Elhage et al., 2021).

**Analyzing Query-Key (QK) pair** First, we explore the spectral flow of the Query-Key (QK) pair, $Q_h(t)^\top K_h(t)$. This particular element plays a vital role in determining particle communication and provides insights into the extent of their interactions.

Figure 2a displays the dynamics of eigenvalues of $Q_h(t)^\top K_h(t)$ in DIFFEQFORMER model. It exhibits a higher degree of interactions in its last layers, as evidenced by the broader range of

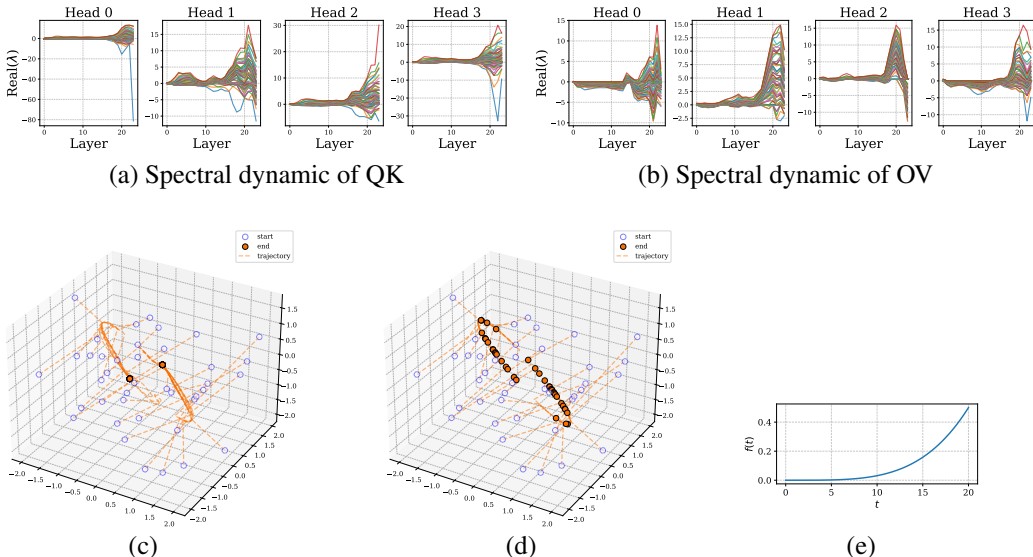

Figure 2: (a-b): Spectral dynamics of QK and OV pairs. (c-d): Trajectory of a sequence consisting of 40 points in 3-dimensional space. (c) Attention-only model with shared weight assumptions as described in Geshkovski et al. (2023a): Clusters emerge. (d) Attention-only model with time-dependent weights of increasing magnitude, inspired by observations from trained DIFFEQ-FORMER: No clusters occur. (e) Plot of a function in our simulation that mimics the magnitude of $Q(t), K(t), V(t)$ over time like in trained DIFFEQFORMER.

eigenvalues in these layers. The eigenvalues of $Q_h(t)^\top K_h(t)$ directly affect the variance of $x_i^\top(t) Q_h(t)^\top K_h(t) x_j(t)$. For instance, if $x, y \sim \mathcal{N}(0, I)$, then $\mathrm{Var}(x^\top A y) = \mathrm{trace}(A^\top A) = \sum_i \lambda_i^2(A)$ where $\lambda_i(A)$ is the $i$-th eigenvalue of $A$. A high variance of $x_i^\top(t) Q_h(t)^\top K_h(t) x_j(t)$ indicates that the softmax of $[x_i^\top(t) Q_h(t)^\top K_h(t) x_j(t)]_{j=1,\ldots,n}$ tends to approach a one-hot vector. This results in sharper attention weights, leading to a more distinct exchange between the $j$-th and $i$-th particles.

To gain a deeper understanding, we make a connection with the work of Geshkovski et al. (2023b). In this study, the authors simplify the model by assuming that $Q(t)$, $K(t)$, and $V(t)$ are all identity matrices, i.e.,

$$f_\beta(x_i(t), x_{[n]}(t), t) = \frac{1}{L_\beta} \sum_{j=1}^n \exp(\beta \langle x_i(t), x_j(t) \rangle) x_j,$$

where $L_\beta = \sum_{j=1}^n \exp(\beta \langle x_i(t), x_j(t) \rangle)$. Here, the parameter $\beta$ serves as a simplification of $Q_h(t)^\top K_h(t)$ in equation 4. The study by Geshkovski et al. (2023b) looks into the interaction energy, and its gradient flows govern the dynamics of $f_\beta$ (Otto, 2001; Jordan et al., 1998; Ambrosio et al., 2005; Villani et al., 2009). Importantly, the parameter $\beta$ directly influences the speed at which the interaction energy decreases or increases. Such interactions in DIFFEQFORMER can be reflected by the eigenvalues of $Q_h(t)^\top K_h(t)$. For example, a wide range of eigenvalues encourage interactions between particles, as they indicate stronger coupling and information exchange within the system.

**Analyzing Output-Value (OV) pair**   While $Q_h(t)^\top K_h(t)$ identifies the source and destination particles of inter-particle communication, the Output-Value (OV) pair represents the component describing how information of the source particle is merged to the destination particle.

Figure 2b illustrates the spectral dynamics of the OV matrix in DIFFEQFORMER. A significant number of heads exhibit positive eigenvalues, which are primarily concentrated near the output layer. It seems to have a correlation with the spectral dynamics of the QK pair.

Consider a communication from a source particle $j$ to a destination particle $i$. The OV matrix has eigenvalues $\{\lambda_k\}_{k=1}^d$ and corresponding eigenvectors $\{v_k\}_{k=1}^d$. We can express the states of

particles $i$ and $j$ at time $t$ as linear combinations of these eigenvectors:

$$x_i(t) = \sum_{k=1}^{d} w_k^i v_k, \quad x_j(t) = \sum_{k=1}^{d} w_k^j v_k,$$

where $\{w_k^i\}_{k=1}^d$ and $\{w_k^j\}_{k=1}^d$ are the respective coefficients. The output of the Euler step for attention can be interpreted as:

$$x_i(t + \Delta t) = \sum_{k=1}^{d} (\lambda_k w_k^j \Delta t + w_k^i) v_k.$$

Here, we focus on the coefficient $\lambda_k w_k^j \Delta t + w_k^i$. The magnitude of $\lambda_k$ determines the influence of the source particle $x_j(t)$ on the output $x_i(t + \Delta t)$. Larger absolute values of $\lambda_k$ result in a stronger influence of the source particle on the destination particle's next state. The sign of $\lambda_k$ decides either positive or negative influence of source particle with respect to the corresponding eigenvector $v_j$.

Drawing connection to the work of Geshkovski et al. (2023a;b), the authors suggest two simplified cases: the *attractive* scenario, where particles exhibit attractions and more likely to follow some leader particles with $V = I_d$; and the *repulsive* scenario with $V = -I_d$, where particles repel each other. We extend this analysis by examining both the sign and magnitude of eigenvalues $\lambda_k$ as shown in Figure 2a and 2b. This means that DIFFEQFORMER shows both attractive and repulsive behaviors, with attraction being more dominant in some attention heads near the last layers.

In comparison, the approach presented by Elhage et al. (2021) characterizes OV pairs as exhibiting copying behavior when the eigenvalues of the OV pair feature numerous positive values. This copying behavior conceptually is similar to the attractive scenario in Geshkovski et al. (2023a;b). In mathematical terms, $V = I_d$ means that all eigenvalues are positive and equal to 1. If copying behaviors occur frequently, it indicates a selective focus on specific particles, resembling a situation where only a subset of major particles is highlighted. Note that copying behavior is characterized as a component of *induction heads*, recognized as a key observation associated with in-context learning performance in Elhage et al. (2021); Olsson et al. (2022).

Under the interpretation of Elhage et al. (2021), induction heads perform two key operations: (i) detecting matching patterns through QK pair and (ii) copying values when matches are found through OV pairs. The "matching" operation manifests through attention scores, where positive eigenvalues of QK pair indicate strong attention paths between similar tokens. The "copying" operation is facilitated by the OV pair, where positive eigenvalues suggest effective information propagation. Given our observations in Figure 2a and 2b of both positive eigenvalues in QK pair (supporting pattern matching) and positive eigenvalues in the OV pair (enabling copying), it is highly probable that induction heads occur in the last layer of DIFFEQFORMER.

**Clustering behavior in DIFFEQFORMER** Trained DIFFEQFORMER exhibits a distinct dynamic of QK and OV pairs characterized by increasing magnitudes of eigenvalues over time, with a notable peak observed near the final layers. This dynamic contrasts with the assumptions put forth by Geshkovski et al. (2023a), who posited the emergence of clustering behavior at limit. Given this unique dynamic, it is crucial to investigate potential clustering behavior in our DIFFEQFORMER. To this end, we simulate ODE trajectories for attention-only model with the simplified dynamic based on the trained DIFFEQFORMER (refer to Appendix C.8 for details). As illustrated in Figure 2d, our findings indicate an absence of emergent clusters for a case of time-dependent weights. This contrasts with the cluster emergence observed in the weight-sharing case, shown in Figure 2c. This outcome aligns with the practical considerations of autoregressive language modeling tasks with next-token prediction, where clustered outputs would be counterintuitive. This shows that a discrepancy between the theoretical assumptions in existing studies and the empirical properties exhibited by our DIFFEQFORMER models. Therefore, it is necessary to have future rigorous investigation into this gap.

**Remark** Due to the continuity of weights within our model, the dynamics of eigenvalues exhibit the continuity property, as discussed in previous works (Wilkinson, 1965; Hoffman & Wielandt, 2003). Consequently, a *smooth* transition of spectral information between layers can be observed in our analysis of QK and OV pairs. In contrast, the spectral flow in the case of vanilla transformers do not exhibit clear patterns or behaviors in their dynamics, making it challenging to identify induction heads (see Appendix E).

BIGBANG is one of those musical entities that transcends language. It's one of those rare groups that both innovates and defines the direction a genre takes. Covering a sound that includes hip hop, R&B and electronic dance, BIGBANG and

(a)

SEOUL (Reuters) - South Korean Lee Sedol won his first match against a computer program developed by a Google subsidiary on Sunday in the ancient board game Go, denying a clean sweep for the artificial intelligence in a five-match series. The world's top Go player Lee Sedol reviews the match after the fourth

(b)

Figure 3: Lyapunov exponent values represent the sensitivity of previous words to the next word. Higher values correspond to more intense highlighting in red. (a) The next word is ***its***. (b) The next word is ***match***.

## 5.2 SENSITIVITY ANALYSIS WITH LYAPUNOV EXPONENTS

Next, we utilize the Lyapunov exponent to assess the sensitivity of the inputs of DIFFEQFORMER. Lyapunov exponents can provide valuable insights into the behavior of transformer models by quantifying how small changes in the input may impact the outputs. Previous studies have utilized Lyapunov exponents for understanding neural networks, primarily focusing on recurrent neural networks (RNNs) (Vogt et al., 2022; Storm et al., 2023; Engelken et al., 2023; Engelken, 2023). In another example, Gilpin (2021) demonstrates how Lyapunov exponents can enhance the interpretability of time series data (see Appendix B for a brief review).

The main challenge in extracting Lyapunov exponents for the entire DIFFEQFORMER system stems from the high dimensionality of its states. Calculating Lyapunov exponents requires computing the spectrum or QR decomposition of Jacobian matrices, which can be computationally infeasible for DIFFEQFORMER[2].

In this context, our objective is to investigate the token-level sensitivity of transformers. Specifically, we aim to understand how the changes of the $i$-th token in a given input sentence influence the $j$-th token of its corresponding output. To achieve this, we intend to monitor the evolution of the tangent vectors associated with the $i$-th input particle and $j$-th output particle, similar to the formulation in equation 8, using the following ODE:

$$\dot{Y}_{ij}(t) = J_{ij}(t)Y_{ij}(t), \tag{7}$$

with $Y_{ij}(0) = I_d$ and $J_{ij}(t) = \partial_{x_i} f(x_j(t), x_{[n]}(t)) + \partial_{x_i} g(x_j(t))$ is the Jacobian of the vector field in equation 4. The Jacobian $J_{ij}$ comprises two components: the attention information and the change between input and output. Consequently, employing Lyapunov exponents can provide a more principled approach to understanding the behavior of transformer models compared to solely using attention matrices, which have limitations as discussed by Jain & Wallace (2019) (see further discussion in Appendix B).

Our approach may share similarities with conditional Lyapunov exponents, which analyze the stability of specific subsystems within a larger system (Pecora et al., 1997). Like these exponents, our sensitivity values provide insights into the stability of specific tokens within the transformer model.

Figure 3 showcases how our approach quantifies sensitivity for predicting the next words using Lyapunov exponents. In these examples, the obtained Lyapunov exponents can explain the case of possessive pronouns. For instance, in the BIGBANG example, the next word "*its*" corresponds to "*BIGBANG*", which is highlighted by the Lyapunov exponent. Similarly, in the Alpha Go article example, the obtained Lyapunov exponent can highlight the next word "*match*" and the order preposition "*after*." This analysis offers potential avenues for interpreting the inner workings of transformers, building upon prior efforts in attention interpretability (Jain & Wallace, 2019; Wiegreffe & Pinter, 2019; Chefer et al., 2021; Ali et al., 2022). Further results and examples of this analysis, including comparisons with attention-based explanations can be found in Appendix F.

## 6 EXPERIMENTAL RESULTS

This section presents an empirical comparison of our proposed model DIFFEQFORMER and various baseline models in the context of language modeling tasks. The implementation of our model

---

[2]For example, consider a model taking 1000 tokens with an embedding dimension of 768 as input; the dimension of its state would be $768,000$.

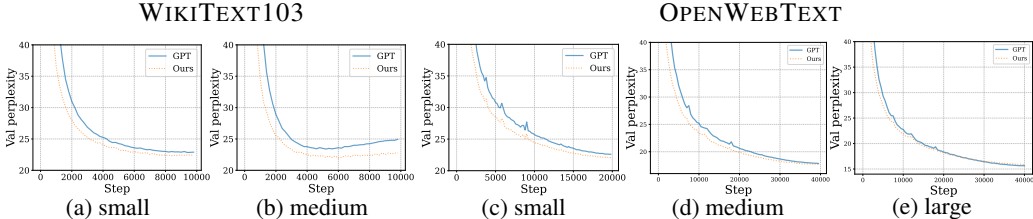

Figure 4: Validation Perplexity of DIFFEQFORMER in comparison with GPT models. (a,b) Results on WIKITEXT103 dataset in two architecture settings. (c, d, e) Results on OPENWEBTEXT dataset on three architecture settings.

and baseline models is built on JAX (Bradbury et al., 2018), utilizing an ecosystem that includes Equinox (Kidger & Garcia, 2021), Haliax, and the Levanter framework (Hall et al., 2023). Our implementation is available at `https://github.com/SDML-KU/qkvflow`.

## 6.1 LANGUAGE MODELING

We evaluate our model DIFFEQFORMER, in comparison to baselines on both OPENWEB-TEXT (Gokaslan et al., 2019) and WIKITEXT103 (Merity et al., 2016) for the autoregressive modeling task. The main evaluation metric is perplexity, a commonly used measure in this task.

**Baselines** We explore various baselines, with the primary one being decoder-only transformers including GPT (Radford, 2018; Brown et al., 2020) and Llama (Touvron et al., 2023). We extend our investigation to a broader range of baselines for the Wikitext dataset, incorporating techniques such as weight-sharing (Zhong et al., 2022). However, on the OPENWEBTEXT dataset, we exclusively compare our model against the GPT model.

**Setting** We consider model settings including: GPT-medium, GPT-small, GPT-large, and Llama-1B (see Table 3 in Appendix for details). We ensure uniformity in terms of model configuration between DIFFEQFORMER and baselines, including the depth of the models, the dimension of token embeddings, the number of heads, and other hyperparameters.

Table 1: Perplexity on WIKITEXT103 data set.

| Model | Perplexity |
|---|---|
| GPT-small | 22.87 |
| Shared parameter | 36.27 |
| Our model-small | **22.84** |
| GPT-medium | 23.38* |
| Shared parameter | 32.95 |
| Our model-medium | **21.94** |

Table 2: Perplexity on OPENWEBTEXT dataset.

| Model | Perplexity |
|---|---|
| GPT-small | 22.60 |
| Our model-small | **22.06** |
| GPT-medium | 17.85 |
| Our model-medium | **17.67** |
| GPT-large | 15.64 |
| Our model-large | **15.43** |
| Llama-1B | 10.17 |
| Our Llama-1B | **9.68** |

Table 1 shows the perplexity across different models on WIKITEXT103 dataset. Our model achieves strong performance, outperforming GPT models while exhibiting less overfitting. Notably, GPT-medium shows signs of overfitting on this data (see Figure 4). This can be attributed to an increase in model parameters without a corresponding increase in data (Kaplan et al., 2020). Though the shared-weight approach excels in masked language modeling (Lan et al., 2020), it falls short in autoregressive language modeling tasks in this experiment.

Table 2 demonstrates that our model consistently outperforms all baselines on the OPENWEBTEXT dataset, with particularly substantial improvements over the Llama-1B baseline (see Appendix C.6).

**Downstream evaluation** We evaluate DIFFEQFORMER against GPT-large and Llama-1B using the `lm-evaluation-harness` framework (Gao et al., 2024), following the experimental protocol established in Biderman et al. (2023). Our evaluation encompasses various downstream tasks in both zero-shot and few-shot settings. While DIFFEQFORMER demonstrates comparable performance to the baselines across most tasks, it achieves statistically significant improvements on reading compre-

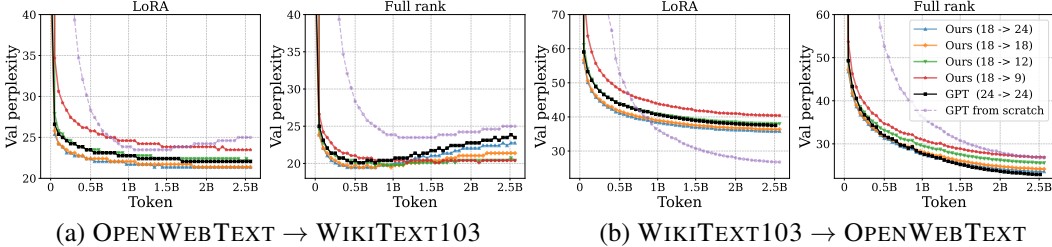

(a) OPENWEBTEXT → WIKITEXT103          (b) WIKITEXT103 → OPENWEBTEXT

Figure 5: Finetune validation perplexity across two settings: (a) OPENWEBTEXT → WIKI-TEXT103; (b) WIKITEXT103 → OPENWEBTEXT. All models are pretrained with 18 function evaluations (or layers) and finetune in different settings of function evaluations 9, 12, 18, 24 with LoRA and full-rank finetune. We compare these with the baseline of corresponding GPT model pretrained and finetune with 24 layers which is on much expensive computation both pretrain and finetune.

hension tasks (Lambada OpenAI and Lambada Standard) in both zero-shot and five-shot evaluations. Detailed results are presented in Appendix C.7.

## 6.2 COMPUTATIONAL ADAPTABILITY FOR FINE-TUNE

Modeling transformers using differential equations enables adaptive computation during fine-tuning through the adjustment of ODE solver step sizes. By utilizing a fixed step size, we populate the weights of DIFFEQFORMER over time steps to obtain discrete-layer pretrained models. Now, we can apply any fine-tuning techiques such as LoRA (Hu et al., 2021).

**Finetune settings** The pretrained model undergoes training on one dataset before evaluation on novel data. We examine two scenarios: pretraining on WIKITEXT103 followed by fine-tuning on OPENWEBTEXT (WIKITEXT103 → OPENWEBTEXT), and pretraining on OPENWEBTEXT followed by fine-tuning on WIKITEXT103 (OPENWEBTEXT → WIKITEXT103). Both LoRA fine-tuning and full-rank weight fine-tuning approaches are employed.

**Model setting** We trained DIFFEQFORMER models with 18 time steps (layers). Fine-tuning was conducted on models with varying time steps: 9, 12, 18, and 24. For instance, pretraining with 18 time steps followed by fine-tuning with 9 time steps is denoted as $18 \rightarrow 9$. A baseline GPT model with 24 layers was used for both training and fine-tuning.

**Result** Figure 5 shows the perplexity evaluated on validation datasets. Our models demonstrate superior performance compared to the GPT baseline across configurations $18 \rightarrow \{12, 18, 24\}$ under LoRA fine-tuning. It is noteworthy that the baseline model was trained and fine-tuned with greater computational resources (24 layers). In the case of full-rank weight fine-tuning (or continued training on new data), our models ($18 \rightarrow 24$) exhibit comparable performance to the baseline, despite initial training with only 18 time steps. Furthermore, the flexibility in fine-tuning allows our models to mitigate overfitting by employing fewer time steps. This is particularly evident in the case of the WIKITEXT103 dataset, where our models ($18 \rightarrow \{9, 12\}$) are less prone to overfitting during the full-rank weight fine-tuning process.

## 7 CONCLUSION

This paper proposes a novel method for formulating transformers as neural ODEs, revealing interesting characteristics within the models. We posit that the findings presented herein can provide theorists with foundation for developing ODE-based transformers in their theoretical frameworks. At the same time, practitioners may explore the potential of integrating ODE approaches into transformers. However, there remain important areas for future work such as improving memory efficiency at larger scales, exploring advanced ODE solvers (McCallum & Foster, 2025), examining the robustness of the proposed model against adversarial attacks (Huang et al., 2021) and extending to various aspects including stochastic differential equation paradigms (Tzen & Raginsky, 2019; Tong et al., 2022) and alternative architectures (Tong et al., 2023).

## REPRODUCIBILITY STATEMENT

The source code of this work is available at `https://github.com/SDML-KU/qkvflow`. We built upon the Levanter library (Hall et al., 2023), which manages Pseudorandom Number Generator (PRNG) states in JAX to ensure bitwise determinism in both data processing and the training process. All hyperparameters are detailed in the Appendix, and the corresponding configuration files can be found in the source code repository.

## ACKNOWLEDGMENTS

This work was supported by Institute of Information & communications Technology Planning & Evaluation (IITP) grants funded by the Korea government(MSIT) (No. RS-2019-II190079, Artificial Intelligence Graduate School Program(Korea University); No. RS-2019-II190075, Artificial Intelligence Graduate School Program (KAIST); No. RS-2022-II220984, Development of Artificial Intelligence Technology for Personalized Plug-and-Play Explanation and Verification of Explanation; No. RS-2024-00457882, AI Research Hub Project) and the New Faculty Settlement Research Fund by Korea University. This work was supported by the New Faculty Settlement Research Fund by Korea University and Artificial intelligence industrial convergence cluster development project funded by the Ministry of Science and ICT(MSIT, Korea) & Gwangju Metropolitan City. We also acknowledge the Google Cloud Research Credit Program for their support during the early development of this work.

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

## A    DETAIL ON TIME-DEPENDENT WEIGHTS

In the main text, we define the time-dependent weights as:
$$W(t) = \text{Proj}(\text{MLP}(\text{Sinusoidal}(t))).$$
Each component will be explained further as follows. The temporal embedding is a concatenation of input time $t$ and sine and cosine of the projection of $t$ in higher dimensions.
$$\text{Sinusoidal(t)} = \text{Concat}(t, \sin(wt), \cos(wt)).$$
The multilayer perceptron is a two-layer neural network with SiLU activation. The projection layer is a linear layer with a reshape function. In our experiment, we set the weight vector $w \in \mathbb{R}^{128}$ as
$$w = \left[ -\log(10^4)\frac{i}{128} \right]_{i=0}^{127}.$$
The hidden layer dimension of the MLP block is set to the embedding dimension $d_{\text{emb}}$.

Note that our models use time-dependent weights for layer normalization, with time embedding serving as input.

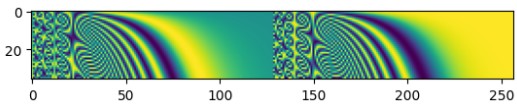

Figure 6: Visualization of the output of Sinusoidal function.

## B    FURTHER DETAILS ON LYAPUNOV EXPONENT

### B.1    BRIEF REVIEW ON LYAPUNOV EXPONENT

In essence, the Lyapunov exponent of a dynamical system quantifies the trajectory separation rate. For a dynamical system described by $\dot{x}(t) = f(x(t))$, consider two trajectories $x^{(1)}(t)$ and $x^{(2)}(t)$. The Lyapunov exponent $\lambda$ roughly captures the separation at time $t$ in relation to the initial separation, expressed as $|x^{(1)}(t) - x^{(2)}(t)| \approx \exp(\lambda t)|x^{(1)}(0) - x^{(2)}(0)|$.

In practice, obtaining the spectrum of Lyapunov involves computing the eigenvalues of matrix $\frac{1}{2t}Y(t)Y^\top(t)$ at the limit where $Y(t)$ satisfies
$$\dot{Y}(t) = J(t)Y(t), \quad Y(0) = I, \tag{8}$$
where $J(t)$ represents the Jacobian matrix of $f(x)$ with respect to $x(t)$. This Jacobian matrix describes how the tangent vectors associated with the given trajectory evolve. The numerical method to compute the Lyapunov exponent can be found in Wolf et al. (1985); Holzfuss & Parlitz (1991); Datseris (2018).

### B.2    DERIVATION OF JACOBIAN TERM IN COMPUTING LYAPUNOV EXPONENT

Consider the case that we want to measure the sensitivity of the $i$-th token to the $j$-th token. We need to extract the Jacobian $J_{ij}$ of the vector field in Equation equation 7.

To make our notation more readable, we rephrase our goal as measure sensitivity of the in-th position to the out-th position. We simplify the multihead setting to one-head setting.

We also denote
$$q_i = Q(t)x_i(t), \quad k_i = K(t)x_i(t), \quad v_i = V(t)x_i(t).$$

The first component of the Jacobian $J_{\text{in,out}}$ now is $\partial_{x_{\text{in}}} f(x_{\text{out}}, x_{[n]})$ and is computed as

$$\partial_{x_{\text{in}}} f(x_{\text{out}}, x_{[n]}) = \begin{bmatrix} \frac{\partial}{\partial q_{\text{in}}} f(x_{\text{out}}, x_{[n]}) \\ \frac{\partial}{\partial k_{\text{in}}} f(x_{\text{out}}, x_{[n]}) \\ \frac{\partial}{\partial v_{\text{in}}} f(x_{\text{out}}, x_{[n]}) \end{bmatrix} \begin{bmatrix} \frac{\partial q_{\text{in}}}{\partial x_{\text{in}}}, \frac{\partial k_{\text{in}}}{\partial x_{\text{in}}}, \frac{\partial v_{\text{in}}}{\partial x_{\text{in}}} \end{bmatrix}.$$

The individual terms in the above equation are computed as

$$\frac{\partial}{\partial q_{\text{in}}} f(x_{\text{out}}, x_{[n]}) = \frac{\partial}{\partial q_{\text{in}}} \sum_{j=1}^{n} \frac{\exp(q_{\text{out}}^{\top} k_j)}{L_{\text{out}}} v_j = 0,$$

$$\frac{\partial}{\partial v_{\text{in}}} f(x_{\text{out}}, x_{[n]}) = \frac{\partial}{\partial v_{\text{in}}} \sum_{j=1}^{n} \frac{\exp(q_{\text{out}}^{\top} k_j)}{L_{\text{out}}} v_j = \frac{\exp(q_{\text{out}}^{\top} k_{\text{in}})}{L_{\text{out}}} I_d = \color{red}{A_{\text{in}\to\text{out}}} I_d,$$

$$\frac{\partial}{\partial k_{\text{in}}} f(x_{\text{out}}, x_{[n]}) = \frac{\partial}{\partial k_{\text{in}}} \sum_{j=1}^{n} \frac{\exp(q_{\text{out}}^{\top} k_j)}{L_{\text{out}}} v_j = \frac{\exp(q_{\text{out}}^{\top} k_{\text{in}})}{L_{\text{out}}} v_{\text{in}} q_{\text{out}}^{\top} - \sum_{j=1}^{n} \frac{\exp(q_{\text{out}}^{\top} k_j)}{L_{\text{out}}} \frac{\exp(q_{\text{out}}^{\top} k_{\text{in}})}{L_{\text{out}}} v_j q_{\text{out}}$$

$$= A_{\text{in}\to\text{out}} [v_{\text{in}} - \sum_{j} A_{j\to\text{out}} v_j] q_{\text{out}}^{\top}$$

$$= A_{\text{in}\to\text{out}} [v_{\text{in}} - f(x_{\text{out}}, x_{[n]})] q_{\text{out}}^{\top}.$$

Here, we denote the attention $A_{i\to j} = \frac{\exp(q_j^{\top} k_i)}{L_j}$ with $L_j = \sum_{\ell} \exp(q_j^{\top} k_{\ell})$.

Also, we have

$$\frac{\partial q_{\text{in}}}{\partial x_{\text{in}}} = Q(t), \quad \frac{\partial k_{\text{in}}}{\partial x_{\text{in}}} = K(t), \quad \frac{\partial v_{\text{in}}}{\partial x_{\text{in}}} = V(t).$$

The second component in the Jacobian $J_{\text{in,out}}$ is $\partial_{x_{\text{in}}} g(x_{\text{out}})$ and can be computed easily.

**Discussion** The Jacobian $J_{\text{in,out}}$ comprises two components: the attention information represented by $\color{red}{A_{\text{in}\to\text{out}}}$ and the change between input and output represented by $\color{blue}{v_{\text{in}} - f(x_{\text{out}}, x_{[n]})}$. This approach differs from heuristic methods that utilize attention matrices to explain transformers, which have been criticized for their limitations (Jain & Wallace, 2019). By incorporating both attention information and input-output changes, our method provides a more principled approach to understanding the behavior of transformer models.

## C  EXPERIMENTS

Here, we provide details about data and experiment set up, e.g., hyperparameters, optimizers.

### C.1  WIKITEXT DATASET

We utilized the Wikitext dataset, accessible via the Hugging Face dataset ID: dlwh/wikitext_103_detokenized.

### C.2  OPENWEBTEXT DATASET

Following the configuration from Dao et al. (2022), we partitioned the entire OpenWebText dataset into training and testing sets. The test set corresponds to 0.0005 of the entire dataset.

### C.3  MODEL SPECIFICATION

The details of model specification of GPT-small, GPT-medium, GPT-large and Llama-1B are described in Table 3.

| Specification | GPT-small | GPT-medium | GPT-large | Llama-1B |
|---|---|---|---|---|
| Parameters | 117M | 345M | 762M | 1213M |
| Layers | 12 | 24 | 36 | 16 |
| Hidden Size | 768 | 1024 | 1280 | 2048 |
| Attention Heads | 12 | 16 | 20 | 16 |

Table 3: Specifications for GPT-small, GPT-medium, GPT-large models, and Llama-1B.

### C.4 HYPERPARAMETER SETTING

All models were trained using the Adam optimizer, along with a weight decay of 0.1 and dropout rate of 0.1. For GPT-small and GPT-medium models, we used Adam's default hyperparameters ($\beta_1 = 0.9, \beta_2 = 0.999$). Based on our findings in Section C.5, we modified these parameters for GPT-large and Llama-1B models, setting $\beta_1 = 0.9, \beta_2 = 0.95$. The number of warm-up steps was set to 1% of the total training steps. Additionally, the cosine learning rate schedule is used. The ratio between the minimum learning rate and the base learning rate was fixed at 0.1.

We mainly use two A100 80GB GPUs for training our models. However, owing to hardware constraints, there are variations in the batch sizes used for training GPT-large, GPT-medium and GPT-small on the OPENWEBTEXT dataset. Specifically, the batch size for GPT-medium and GPT-large is set to 256, while the batch size for GPT-small is configured at 512. Meanwhile, the batch size for training on the WIKITEXT103 dataset is set at 256 for all models. For experiments with the Llama-1B configuration, training was conducted using eight NVIDIA H100 GPUs, each with 80GB of memory.

**Backpropagation** We employ the optimal online checkpointing technique (Stumm & Walther, 2010) rather than opting for the optimize-then-discretize approach or the adjoint method, as outlined in Chen et al. (2018). The latter methods may introduce error accumulation, leading to instability during training. For a thorough discussion of the advantages and disadvantages of these techniques in backpropagation for neural ODEs, one can refer to (Kidger, 2022, Chapter 5). Notably, this approach shares similarities with the techniques used in training deep neural networks (Chen et al., 2016; Gruslys et al., 2016; rey Zweig & Padmanabhan, 2000).

### C.5 ON ADAM'S $\beta_2$ PARAMETER

Since DIFFEQFORMER's weights are generated by time-dependent weight functions, its parameterization differs from GPT models, leading to distinct loss landscapes and optimizer behaviors. In practice, we found that setting the Adam optimizer parameter $\beta_2 = 0.95$ improves optimization stability and convergence for DIFFEQFORMER compared to the standard value of 0.999.

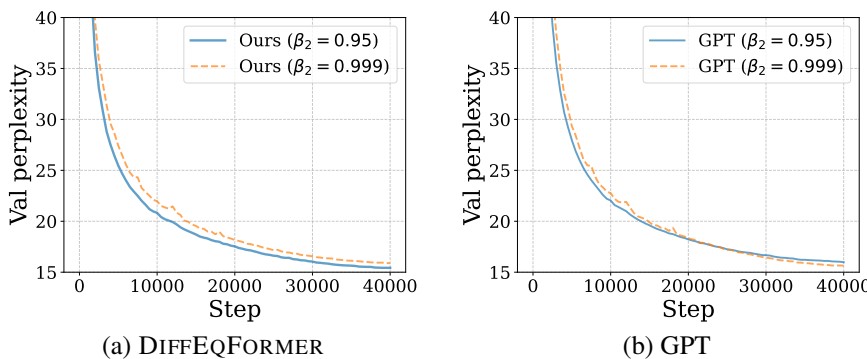

(a) DIFFEQFORMER        (b) GPT

Figure 7: The effect of modifying Adam's $\beta_2$ parameter to 0.95. (a) DIFFEQFORMER shows significant performance improvement with $\beta_2 = 0.95$ (from 15.89 to 15.43). In contrast, while GPT initially converges faster with $\beta_2 = 0.95$, its final performance is inferior to that achieved with $\beta_2 = 0.999$. Both models exhibit more stable and smoother training curves with decreased $\beta_2$ values.

When comparing DIFFEQFORMER with GPT-large on the OPENWEBTEXT dataset, using $\beta_2 = 0.95$ reduced our model's validation perplexity from 15.89 to 15.43, surpassing GPT baselines. In contrast, GPT models perform better with $\beta_2 = 0.999$, as shown in Figure 7. We observe that lower $\beta_2$ values generally produce smoother learning curves in both models.

We hypothesize that this behavior arises because the parameters of time-dependent weights (the MLP block in $W(t)$'s architecture) require rapid adaptation to generate flexible weights. Since

higher $\beta_2$ values maintain longer memory of past squared gradients, a lower $\beta_2$ enables quicker adaptation of time-dependent weights by reducing the influence of historical gradients. This can explain the improvement for our model shown in Figure 7.

## C.6 OTHER BASELINE ARCHITECTURES

We conduct additional experiments to evaluate the adaptability of DIFFEQFORMER across different architectures. Specifically, we implement a LLaMA-based version of DIFFEQFORMER following the architecture proposed by Touvron et al. (2023). The key architectural differences between LLaMA-based and GPT-based implementations include: (1) the use of rotary positional embeddings (RoPE) instead of absolute positional embeddings, (2) RMSNorm for layer normalization rather than the standard LayerNorm, and (3) SwiGLU activation functions in feed-forward blocks instead of ReLU. Importantly, these architectural modifications do not alter the core principles and effectiveness of DIFFEQFORMER's design.

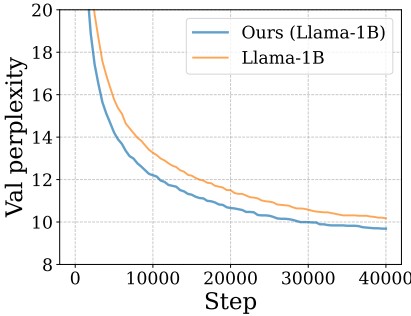

Figure 8: Comparison of validation perplexity between our model and Llama-1B, both trained with Adam optimizer ($\beta_1 = 0.9, \beta_2 = 0.95$).

Figure 8 compares the validation perplexity between our Llama-based DIFFEQFORMER and the baseline Llama-1B on the OPENWEBTEXT dataset. Our model consistently achieves lower perplexity, demonstrating DIFFEQFORMER's effectiveness when implemented with modern architectures like Llama.

## C.7 DOWNSTREAM TASK EVALUATION

We evaluate the performance of DIFFEQFORMER and baselines in the setting of GPT-large and Llama-1B. We use the framework `lm-evaluation-harness` (Gao et al., 2024) to evaluation the pre-trained models. The performance is assessed with the tasks having the similar setup with Biderman et al. (2023). Figure 9, Figure 10, and Table 4 show the performance compared against the GPT-large baseline across different tasks in two settings: zero-shot and five-shot. Figure 11, Figure 12, and Table 5 show the performance compared against the Llama-1B baseline across different tasks in two settings: zero-shot and five-shot.

**GPT-large** Our model demonstrates competitive performance against GPT-large across multiple benchmarks. In zero-shot evaluations (Figure 9), we achieve statistically significant improvements on reading comprehension tasks: Lambada OpenAI and Lambada Standard. Additionally, we observe modest but consistent gains on reasoning tasks including LogiQA, PIQA, and SciQ. In five-shot settings (Figure 10), our model outperforms GPT-large on 5 out of 10 tasks, with statistically significant improvements on Lambada Standard and Winogrande. Performance remains comparable on two tasks, while GPT-large shows marginal, though not statistically significant, advantages on two others. These results suggest our neural ODE architecture maintains or exceeds GPT-large's capabilities across diverse language understanding and reasoning tasks.

**Llama-1B** Our model demonstrates notable improvements over Llama-1B in both zero-shot and five-shot settings, as shown in Figures 11 and 12. The improvements are particularly pronounced in

reading comprehension tasks - we observe statistically significant accuracy gains (2% - 4%) on both Lambada OpenAI and Lambada Standard benchmarks. These results suggest our neural ODE-based transformer architecture enhances language understanding capabilities while maintaining competitive performance across other evaluation metrics.

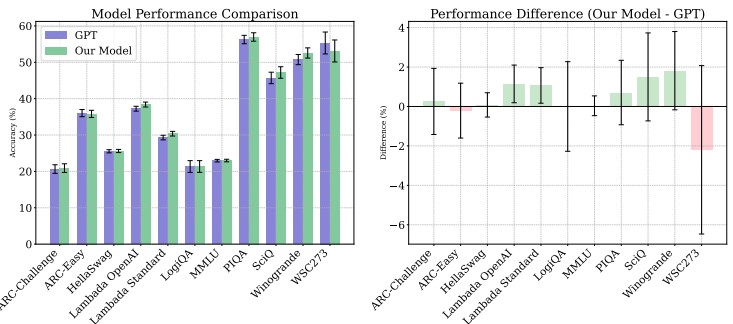

Figure 9: Benchmark score for zero-shot in GPT-large configuration.

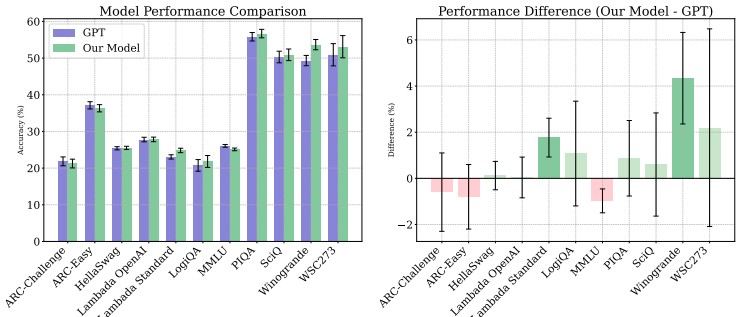

Figure 10: Benchmark score for five-shot GPT-large configuration.

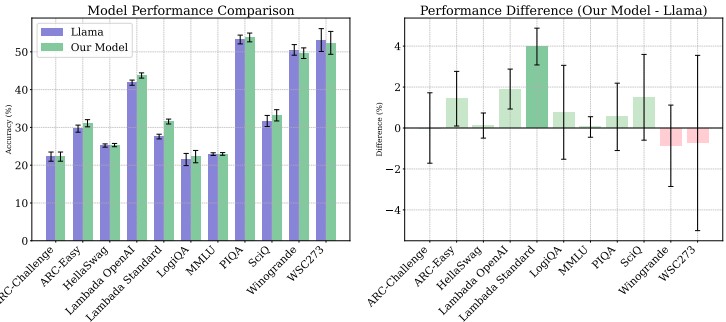

Figure 11: Benchmark score for zero-shot in Llama-1B configuration.

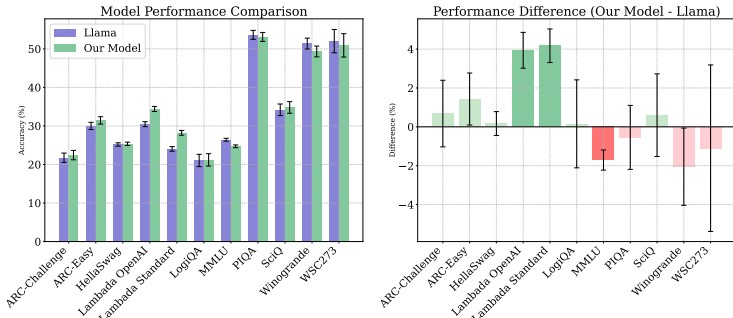

Figure 12: Benchmark score for five-shot in Llama-1B configuration.

| Benchmark | Zero-shot | | Five-shot | |
|---|---|---|---|---|
| | GPT | Our Model | GPT | Our Model |
| ARC-Challenge | 20.65 (1.18) | 20.90 (1.19) | 21.84 (1.21) | 21.25 (1.20) |
| ARC-Easy | 36.03 (0.99) | 35.82 (0.98) | 37.12 (0.99) | 36.32 (0.99) |
| HellaSwag | 25.53 (0.44) | 25.61 (0.44) | 25.40 (0.43) | 25.52 (0.44) |
| Lambada OpenAI | 37.24 (0.67) | 38.39 (0.68) | 27.79 (0.62) | 27.83 (0.62) |
| Lambada Standard | 29.30 (0.63) | 30.37 (0.64) | 23.05 (0.59) | 24.82 (0.60) |
| LogiQA | 21.35 (1.61) | 21.35 (1.61) | 20.74 (1.59) | 21.81 (1.62) |
| MMLU | 22.95 (0.35) | 22.99 (0.35) | 26.08 (0.37) | 25.10 (0.36) |
| PIQA | 56.26 (1.16) | 56.96 (1.16) | 55.82 (1.16) | 56.69 (1.16) |
| SciQ | 45.70 (1.58) | 47.20 (1.58) | 50.30 (1.58) | 50.90 (1.58) |
| Winogrande | 50.75 (1.41) | 52.57 (1.40) | 49.33 (1.41) | 53.67 (1.40) |
| WSC273 | 55.31 (3.01) | 53.11 (3.03) | 50.92 (3.03) | 53.11 (3.03) |

Table 4: Performance comparison between GPT and our model in zero-shot and five-shot settings.

| Benchmark | Zero-shot | | Five-shot | |
|---|---|---|---|---|
| | Llama-1B | Our Model | Llama-1B | Our Model |
| ARC-Challenge | 22.27 (1.22) | 22.27 (1.22) | 21.76 (1.21) | 22.44 (1.22) |
| ARC-Easy | 29.67 (0.94) | 31.10 (0.95) | 30.01 (0.94) | 31.44 (0.95) |
| HellaSwag | 25.20 (0.43) | 25.32 (0.43) | 25.19 (0.43) | 25.36 (0.43) |
| Lambada OpenAI | 41.84 (0.69) | 43.74 (0.69) | 30.47 (0.64) | 34.41 (0.66) |
| Lambada Standard | 27.58 (0.62) | 31.55 (0.65) | 24.04 (0.60) | 28.22 (0.63) |
| LogiQA | 21.51 (1.61) | 22.27 (1.63) | 21.04 (1.60) | 21.20 (1.60) |
| MMLU | 22.93 (0.35) | 22.98 (0.35) | 26.43 (0.37) | 24.72 (0.36) |
| PIQA | 53.26 (1.16) | 53.81 (1.16) | 53.65 (1.16) | 53.10 (1.16) |
| SciQ | 31.70 (1.47) | 33.20 (1.49) | 34.20 (1.50) | 34.80 (1.51) |
| Winogrande | 50.51 (1.41) | 49.64 (1.41) | 51.38 (1.40) | 49.33 (1.41) |
| WSC273 | 53.11 (3.03) | 52.38 (3.03) | 52.01 (3.03) | 50.92 (3.03) |

Table 5: Performance comparison between Llama and our model in zero-shot and five-shot settings.

## C.8    ON TRAJECTORY SIMULATION

This simulation tries to replicate scenarios following the spectral dynamic of trained DIFFEQ-FORMER: *increasing the magnitude of eigenvalues with peak near the last layer.*

We consider the attention-only model with one head,

$$\dot{x}_i(t) = f(x_i(t), x_{[n]}(t), t), \qquad\qquad t \in [0, T],$$
$$x_i(0) = X_i, \qquad\qquad i = 1, \ldots, n.$$

with

$$f(x_i(t), x_{[n]}(t), t) = \sum_{j=1}^{n} \exp\left(\frac{\langle Q(t)x_i(t), K(t)x_j(t)\rangle}{\sqrt{d}}\right) V(t)x_j(t),$$

The weights are randomly initialized at time $0$ and time-dependent under the following dynamics:

$$Q(t) = K(t) = A_0 f(t) \qquad\qquad A_0^{i,j} \sim \mathcal{N}(0,1),$$
$$V(t) = V_0 f(t) \qquad\qquad V_0^{i,j} \sim \mathcal{N}(0,1).$$

where $f(t)$ controls the magnitude of the matrices over time. We test different configurations of $f(t)$ as polynomials including:

$$f_0(t) = \frac{1}{2}, \qquad\qquad f_3(t) = \frac{1}{2}\left(\frac{t}{T}\right)^3,$$
$$f_1(t) = \frac{1}{2}\frac{t}{T}, \qquad\qquad f_4(t) = \frac{1}{2}\left(\frac{t}{T}\right)^4,$$
$$f_2(t) = \frac{1}{2}\left(\frac{t}{T}\right)^2, \qquad\qquad f_5(t) = \frac{1}{2}\left(1 - \frac{t}{T}\right)^2.$$

In our simulation, we employed the following parameters: a time horizon $T = 20$ and an ODE solver step size $dt = 0.1$, effectively resulting in 200 layers of attention. We generated a sequence of length 40 with dimension 3, using uniform random sampling within the range $[-2, 2]$.

Figure 13 illustrates the trajectories of ODEs given the inputs. We observe that dynamics characterized by functions increasing at rates exceeding linear tend not to form clusters. Moreover, we noted that the output dispersion increases with the polynomial order of the function. For instance, when $f_1$ is defined as a linear function of $t$, the corresponding ODE trajectory exhibits incipient cluster formation. In contrast, when $f_4$ is represented by a fourth-order polynomial, its outputs demonstrate greater scatter.

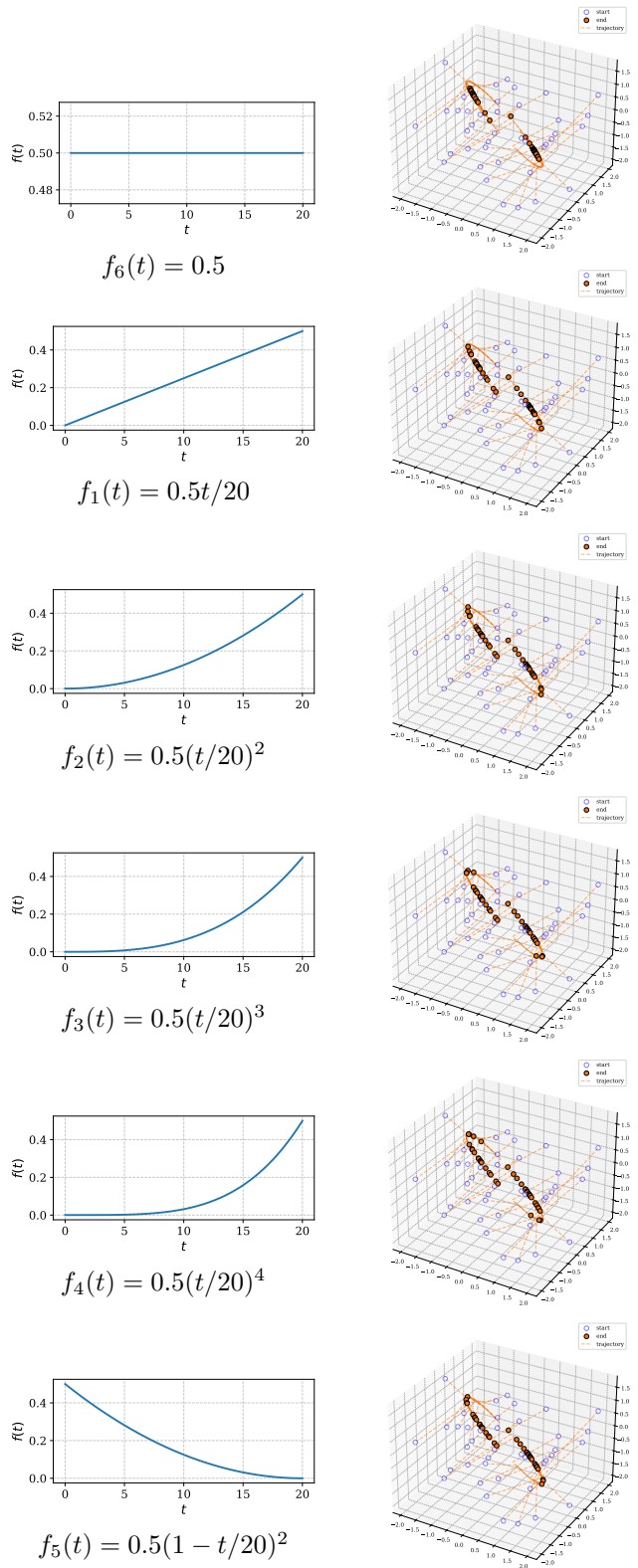

Figure 13: Trajectories of ODE dynamics for various settings of $f(t)$.

# D   ABLATION STUDY

Here we study our model when varying time-embedding dimension. Then we investigate an alternative architecture for time-dependent weights.

## D.1   ON THE DIMENSION OF TIME EMBEDDINGS

In equation 6, $d_{\text{emb}}$ denotes the dimensionality of the time embedding used to derive the time-dependent weight $W(t)$. This hyperparameter directly influences the model's parameter count, with higher values potentially leading to increased model complexity. Our investigation aims to assess the impact of varying $d_{\text{emb}}$ on model performance. Figure 14 illustrates the validation perplexity of DIFFEQFORMER across different $d_{\text{emb}}$ values, benchmarked against the GPT-large model. The results demonstrate a positive correlation between increasing $d_{\text{emb}}$ and improved performance. For models trained on the OPENWEBTEXT dataset, relatively high $d_{\text{emb}}$ values 144 are necessary to achieve optimal performance. Moreover, models with larger hidden dimensions require correspondingly higher $d_{\text{emb}}$ values. We posit that $d_{\text{emb}}$ is related to the concept of intrinsic dimension, as described by Li et al. (2018). This connection stems from the fact that in equation 6, the time-dependent weight $W(t)$ results from projecting a low-dimensional vector of size $d_{\text{emb}}$.

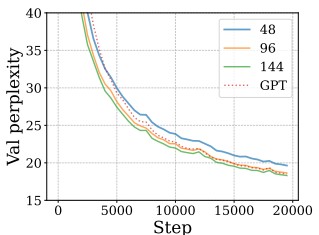

Figure 14: Perplexity of DIF-FEQFORMER with varying $d_{\text{emb}}$.

Figure 15 shows the training throughput between two models. As $d_{\text{emb}}$ increases, a slight decrease in throughput is observed. However, the magnitude of this decrease is notably small, suggesting that DIFFEQFORMER maintains competitive throughput even with higher-dimensional time embedding. Despite DIFFEQFORMER having a significantly higher parameter count compared to the GPT model (see Table 6) due to its time-dependent weights, this has a minimal impact on the overall throughput. The additional computational cost is primarily incurred during the precomputation of the weights prior to the forward pass. The forward pass itself has the same computational cost as that of the GPT model. For the same reason, the GPU usage of DIFFEQFORMER can be similar to that of vanilla transformers. We are able to train DIFFEQFORMER with a medium-sized architecture using two NVIDIA A100 GPUs, despite having 1.8 billion parameters ($d_{\text{emb}} = 144$).

**Inference time**   While learning the time-dependent weights ($W(t)$) in transformers requires both forward and backward passes during training, this process becomes unnecessary during inference. Since the training is complete, we can precompute $W(t)$ for each layer and reuse the computed weights when needed.

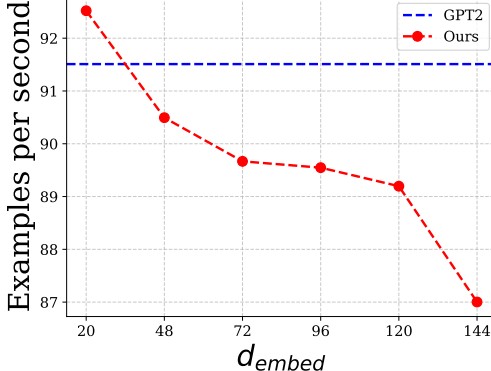

Figure 15: Throughput comparison, measured in examples per second, between GPT and DIFFEQFORMER for different values of the time embedding dimension ($d_{\text{emb}}$).

| $d_{\text{emb}}$ | Number of parameters |
|---|---|
| 20 | 300M |
| 48 | 650M |
| 72 | 960M |
| 96 | 1250M |
| 120 | 1560M |
| 144 | 1860M |

Table 6: Parameter count of DIF-FEQFORMER with medium-sized architecture.

**GPU memory usage** Figure 16 shows the peak GPU memory usage across different $d_{\text{emb}}$ values when training DIFFEQFORMER with GPT-medium architecture at batch size 256. While increasing $d_{\text{emb}}$ expands the model's parameter count, it does not proportionally increase GPU memory usage for multiplicative factors. This is because the time-dependent weights $W(t)$ are computed once per forward pass to generate the weights. Such generate weights then interact with transformer hidden states across batches.

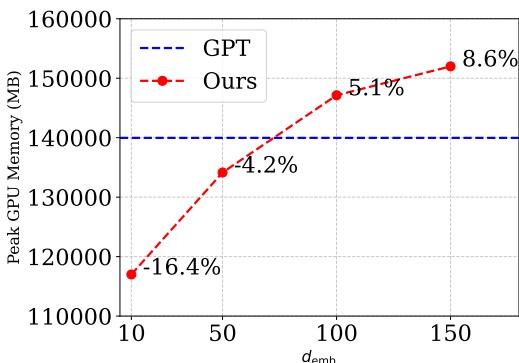

Figure 16: Peak memory of GPU for different settings of $d_{\text{emb}}$. Note that the GPU memory values are obtained after setting JAX environment variable XLA_PYTHON_CLIENT_ALLOCATOR=platform.

### D.2 ALTERNATIVE DESIGN FOR TIME-DEPENDENT WEIGHTS

Figure 18 shows the performance between two designs of time-dependent weights: (i) our model shares only Sinusoidal; (ii) alternative design shares both Sinusoidal and MLP blocks (see Figure 17). We observe the our model outperforms the alternative due to having non-linearity before the projection block.

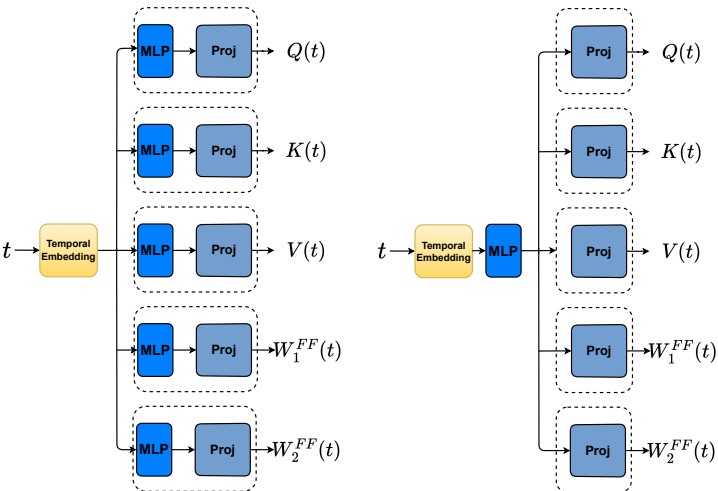

Figure 17: **Left**: Our model's time-dependent weights: sharing Sinusoidal component for all weights $Q(t), K(t), V(t), W_1^{FF}, W^{FF}(t)_2(t)$. **Right**: An alternative architecture shares both Sinusoidal and MLP block.

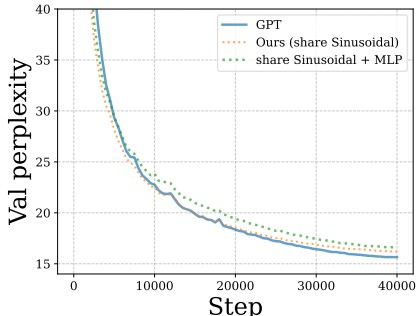

Figure 18

# E    SPECTRAL DYNAMICS

This section presents the spectral dynamics for each dataset and model setting, as described in the main text.

## E.1    ATTENTION BLOCKS

### E.1.1    OPENWEBTEXT DATA

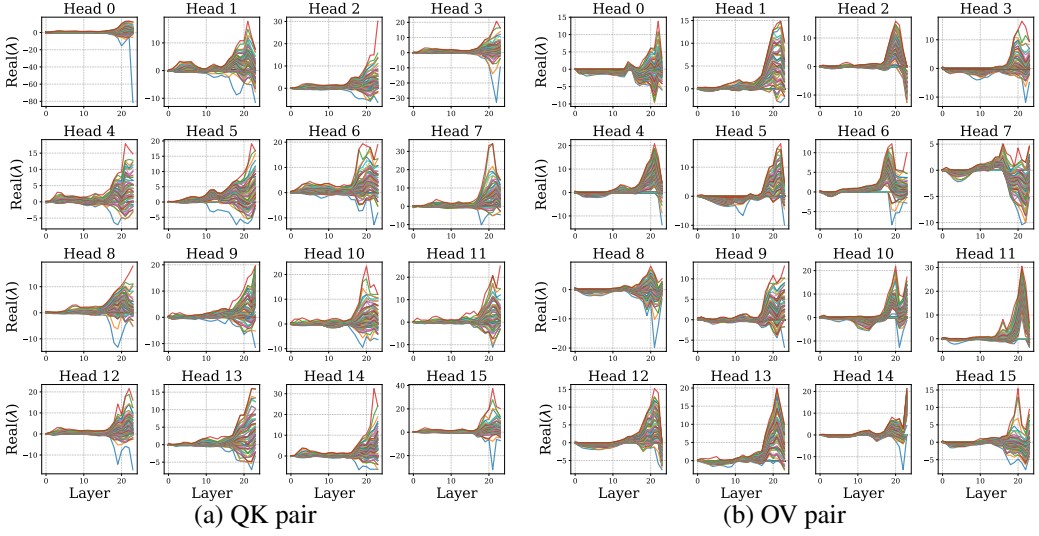

Figure 19: The spectral dynamics of the Query-Key (QK) and Output-Value (OV) pairs in the DIF-FEQFORMER model trained on the OPENWEBTEXT dataset using a medium-sized architecture.

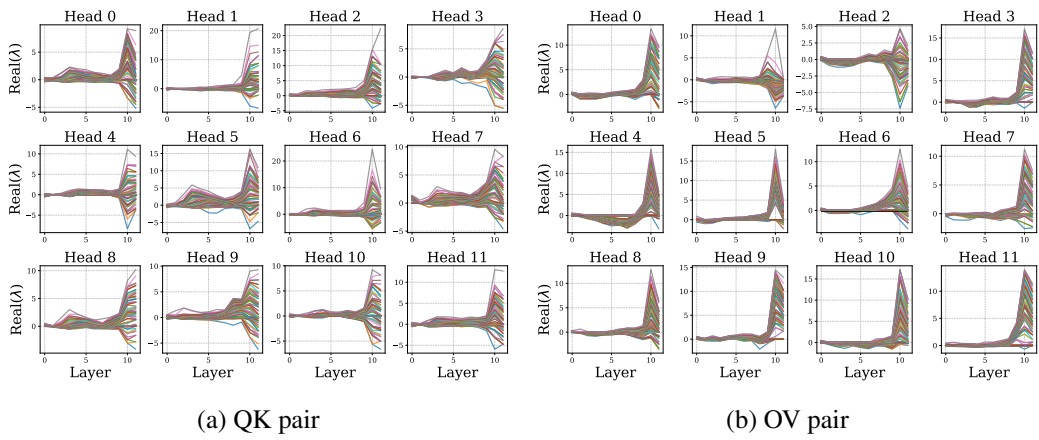

(a) QK pair    (b) OV pair

Figure 20: The spectral dynamics of the Query-Key (QK) and Output-Value (OV) pairs in the DIF-FEQFORMER model trained on the OPENWEBTEXT dataset using a small-sized architecture.

### E.1.2 WIKITEXT103 DATA

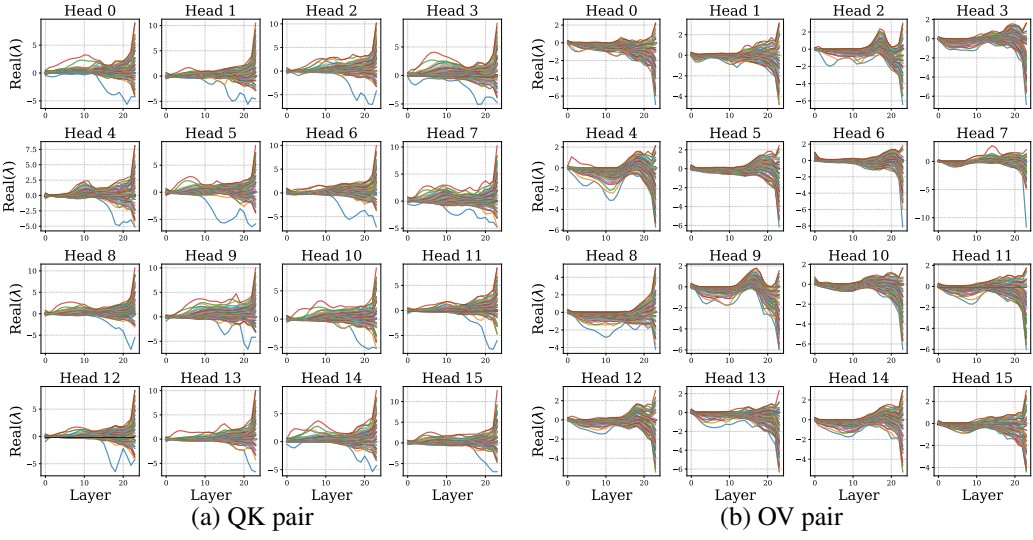

(a) QK pair    (b) OV pair

Figure 21: The spectral dynamics of the Query-Key (QK) and Output-Value (OV) pairs in the DIF-FEQFORMER model trained on the WIKITEXT103 dataset using a medium-sized architecture.

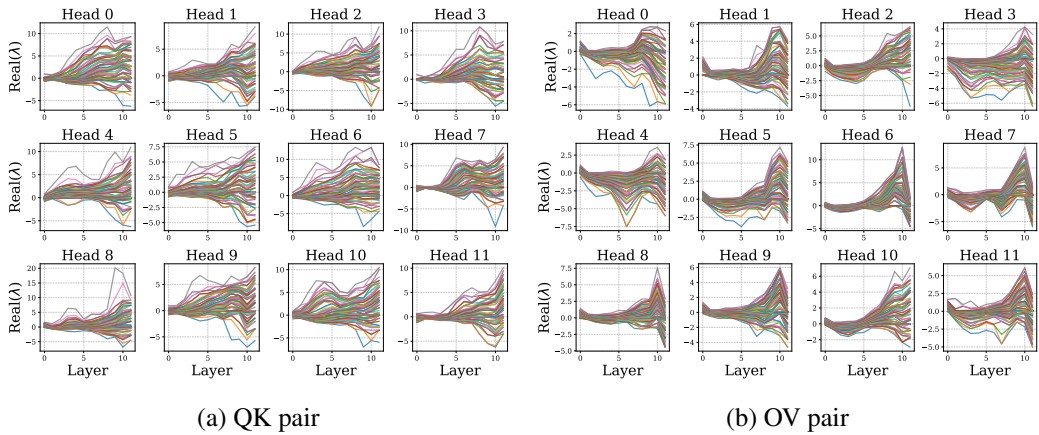

Figure 22: The spectral dynamics of the Query-Key (QK) and Output-Value (OV) pairs in the DIF-FEQFORMER model trained on the WIKITEXT103 dataset using a small-sized architecture.

### E.1.3 VANILLA TRANSFORMER

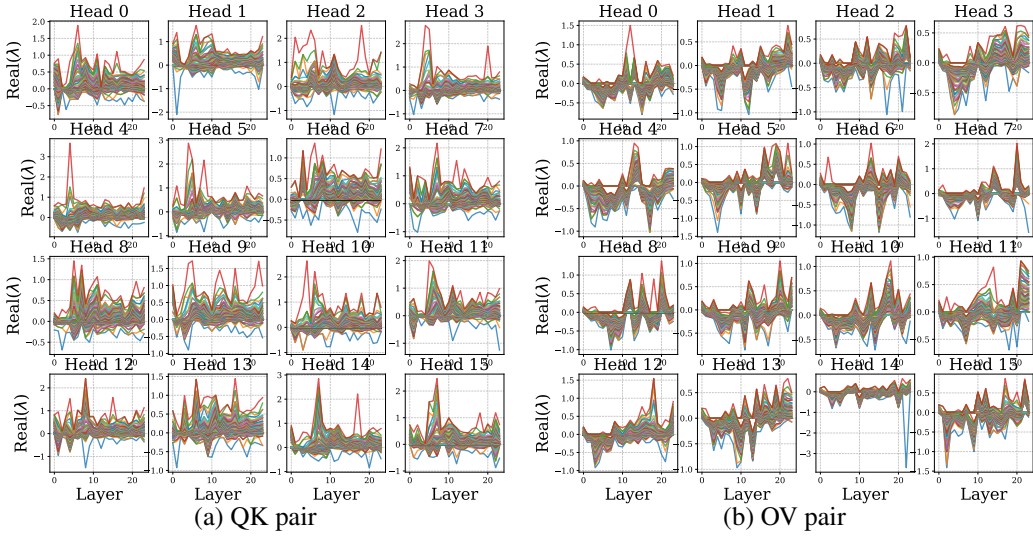

Figure 23: The spectral dynamics of the Query-Key (QK) and Output-Value (OV) pairs in the vanilla transformer model trained on the OPENWEBTEXT dataset using a medium-sized architecture.

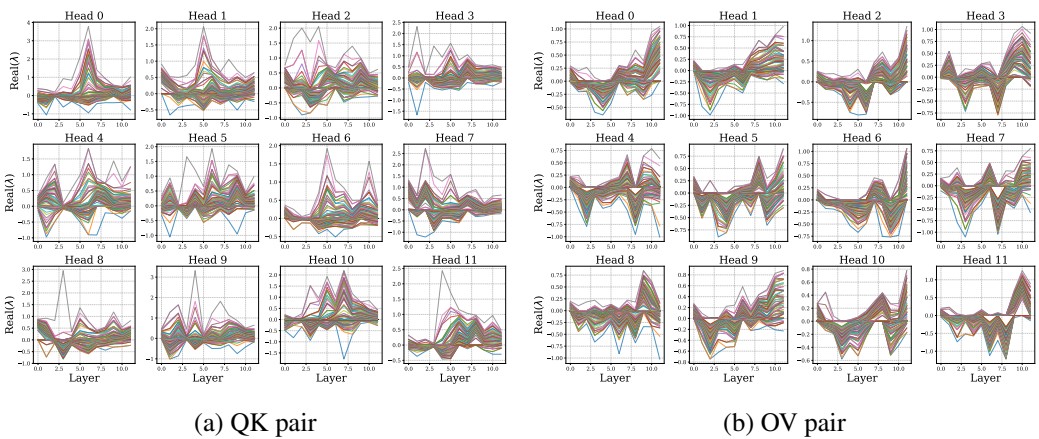

(a) QK pair                                        (b) OV pair

Figure 24: The spectral dynamics of the Query-Key (QK) and Output-Value (OV) pairs in the vanilla transformer model trained on the OPENWEBTEXT dataset using a small-sized architecture.

## F  LYAPUNOV SENSITIVITY EXAMPLES

More examples for Lyapunov sensitivity can be found in the following figures.

(A)  BIGBANG is one of those musical entities that transcends language.
It's one of those rare groups that both innovates and defines the
direction a genre takes. Covering a sound that includes hip hop, R&B
and electronic dance, BIGBANG and

---

(B)  **Average of all heads**

BIGBANG is one of those musical entities that transcends language. It's one of
those rare groups that both innovates and defines the direction a genre takes.
Covering a sound that includes hip hop, R&B and electronic dance, BIGBANG and

**Head 0**

BIGBANG is one of those musical entities that transcends language. It's one of
those rare groups that both innovates and defines the direction a genre takes.
Covering a sound that includes hip hop, R&B and electronic dance, BIGBANG and

**Head 1**

BIGBANG is one of those musical entities that transcends language. It's one of
those rare groups that both innovates and defines the direction a genre takes.
Covering a sound that includes hip hop, R&B and electronic dance, BIGBANG and

**Head 2**

BIGBANG is one of those musical entities that transcends language. It's one of
those rare groups that both innovates and defines the direction a genre takes.
Covering a sound that includes hip hop, R&B and electronic dance, BIGBANG and

**Head 3**

BIGBANG is one of those musical entities that transcends language. It's one of
those rare groups that both innovates and defines the direction a genre takes.
Covering a sound that includes hip hop, R&B and electronic dance, BIGBANG and

**Head 4**

BIGBANG is one of those musical entities that transcends language. It's one of
those rare groups that both innovates and defines the direction a genre takes.
Covering a sound that includes hip hop, R&B and electronic dance, BIGBANG and

Figure 25: The next word is *its* which is the possessive pronoun for BIGBANG. (A) This shows
that the obtained Lyapunov sensitivity can identify the main subject for possessive pronoun. (B)
Analysis of token relationships through attention matrices. While individual attention heads capture
specific aspects of token dependencies, the aggregated attention matrix across all heads does not
fully explain next-token predictions.

(A) North Korea has detained another U.S. citizen, a Korean American professor, bringing to three the number of Americans being held in Pyongyang. The Swedish Embassy in Pyongyang, which represents U.S. interests there because the United States does not have diplomatic relations with

(B)

**Average of all heads**

North Korea has detained another U.S. citizen, a Korean American professor, bringing to three the number of Americans being held in Pyongyang. The Swedish Embassy in Pyongyang, which represents U.S. interests there because the United States does not have diplomatic relations with

**Head 0**

North Korea has detained another U.S. citizen, a Korean American professor, bringing to three the number of Americans being held in Pyongyang. The Swedish Embassy in Pyongyang, which represents U.S. interests there because the United States does not have diplomatic relations with

**Head 1**

North Korea has detained another U.S. citizen, a Korean American professor, bringing to three the number of Americans being held in Pyongyang. The Swedish Embassy in Pyongyang, which represents U.S. interests there because the United States does not have diplomatic relations with

**Head 2**

North Korea has detained another U.S. citizen, a Korean American professor, bringing to three the number of Americans being held in Pyongyang. The Swedish Embassy in Pyongyang, which represents U.S. interests there because the United States does not have diplomatic relations with

**Head 3**

North Korea has detained another U.S. citizen, a Korean American professor, bringing to three the number of Americans being held in Pyongyang. The Swedish Embassy in Pyongyang, which represents U.S. interests there because the United States does not have diplomatic relations with

Figure 26: The next word is **North** in North Korea. (A) We also can see that Lyapunov sensitivity highlights the related words including "North", "Korean", "diplomatic" in this context. (B) Analysis of token relationships through attention matrices. We observe the same behaviors as the previous example.

(A)

Sleep is present and tightly regulated in every vertebrate species in which it has been carefully investigated, but what sleep is for remains a mystery. Sleep is also present in invertebrates, and an extensive analysis in Drosophila melanogaster has shown that sleep in fruit flies shows most of the fundamental features that characterize sleep in mammals. In Drosophila, sleep consists of sustained periods of quiescence associated with an increased arousal threshold. Fly sleep is modulated by several of the same stimulants and hypnotics that affect mammalian

---

(B)

**Average of all heads**

Sleep is present and tightly regulated in every vertebrate species in which it has been carefully investigated, but what sleep is for remains a mystery. Sleep is also present in invertebrates, and an extensive analysis in Drosophila melanogaster has shown that sleep in fruit flies shows most of the fundamental features that characterize sleep in mammals. In Drosophila, sleep consists of sustained periods of quiescence associated with an increased arousal threshold. Fly sleep is modulated by several of the same stimulants and hypnotics that affect mammalian

**Head 0**

Sleep is present and tightly regulated in every vertebrate species in which it has been carefully investigated, but what sleep is for remains a mystery. Sleep is also present in invertebrates, and an extensive analysis in Drosophila melanogaster has shown that sleep in fruit flies shows most of the fundamental features that characterize sleep in mammals. In Drosophila, sleep consists of sustained periods of quiescence associated with an increased arousal threshold. Fly sleep is modulated by several of the same stimulants and hypnotics that affect mammalian

**Head 1**

Sleep is present and tightly regulated in every vertebrate species in which it has been carefully investigated, but what sleep is for remains a mystery. Sleep is also present in invertebrates, and an extensive analysis in Drosophila melanogaster has shown that sleep in fruit flies shows most of the fundamental features that characterize sleep in mammals. In Drosophila, sleep consists of sustained periods of quiescence associated with an increased arousal threshold. Fly sleep is modulated by several of the same stimulants and hypnotics that affect mammalian

**Head 2**

Sleep is present and tightly regulated in every vertebrate species in which it has been carefully investigated, but what sleep is for remains a mystery. Sleep is also present in invertebrates, and an extensive analysis in Drosophila melanogaster has shown that sleep in fruit flies shows most of the fundamental features that characterize sleep in mammals. In Drosophila, sleep consists of sustained periods of quiescence associated with an increased arousal threshold. Fly sleep is modulated by several of the same stimulants and hypnotics that affect mammalian

Figure 27: The next word is *sleep*. (A) The Lyapunov sensitivity can highlight the word "sleep" which is the main topic of the paragraph and appears at the beginning of the sentence. In this case, it may overestimate the sensitivity of the word "mammalian". (B) Analysis of token relationships through attention matrices. We observe the same behaviors as the previous example.

(A)

SEOUL (Reuters) - South Korean Lee Sedol won his first match
against a computer program developed by a Google subsidiary on
Sunday in the ancient board game Go, denying a clean sweep for the
artificial intelligence in a five-match series. The world's top Go player
Lee Sedol reviews the match after the fourth

(B) **Average of all heads**

SEOUL (Reuters) - South Korean Lee Sedol won his first match against a computer program developed by a Google
subsidiary on Sunday in the ancient board game Go, denying a clean sweep for the artificial intelligence in a five-
match series. The world's top Go player Lee Sedol reviews the match after the fourth

**Head 0**

SEOUL (Reuters) - South Korean Lee Sedol won his first match against a computer program developed by a Google
subsidiary on Sunday in the ancient board game Go, denying a clean sweep for the artificial intelligence in a five-
match series. The world's top Go player Lee Sedol reviews the match after the fourth

**Head 1**

SEOUL (Reuters) - South Korean Lee Sedol won his first match against a computer program developed by a Google
subsidiary on Sunday in the ancient board game Go, denying a clean sweep for the artificial intelligence in a five-
match series. The world's top Go player Lee Sedol reviews the match after the fourth

**Head 2**

SEOUL (Reuters) - South Korean Lee Sedol won his first match against a computer program developed by a Google
subsidiary on Sunday in the ancient board game Go, denying a clean sweep for the artificial intelligence in a five-
match series. The world's top Go player Lee Sedol reviews the match after the fourth

**Head 3**

SEOUL (Reuters) - South Korean Lee Sedol won his first match against a computer program developed by a Google
subsidiary on Sunday in the ancient board game Go, denying a clean sweep for the artificial intelligence in a five-
match series. The world's top Go player Lee Sedol reviews the match after the fourth

**Head 4**

SEOUL (Reuters) - South Korean Lee Sedol won his first match against a computer program developed by a Google
subsidiary on Sunday in the ancient board game Go, denying a clean sweep for the artificial intelligence in a five-
match series. The world's top Go player Lee Sedol reviews the match after the fourth

Figure 28: The next word is ***match***. (A) This is an interesting case where the output of Lyapunov
sensitivity highlights the word "match" in "five-match" and the word "after" which indicates the
order of the matches. (B) Analysis of token relationships through attention matrices. We observe
the same behaviors as the previous example.

(A) An ancient Roman merchant vessel has been discovered off the Italian coastline, reportedly in such good condition that much of the food it was carrying might still be intact in its storage jars. 'There are some broken jars around the wreck, but we believe that most of the amphorae inside the ship are still sealed and food-filled," Lt. Col. Francesco Schilardi of the police divers' group told the BBC of the containers. Local fisherman first became aware of the wreck when pieces of pottery began turning up in their

(B) **Average of all heads**

An ancient Roman merchant vessel has been discovered off the Italian coastline, reportedly in such good condition that much of the food it was carrying might still be intact in its storage jars. 'There are some broken jars around the wreck, but we believe that most of the amphorae inside the ship are still sealed and food-filled," Lt. Col. Francesco Schilardi of the police divers' group told the BBC of the containers. Local fisherman first became aware of the wreck when pieces of pottery began turning up in their

**Head 0**

An ancient Roman merchant vessel has been discovered off the Italian coastline, reportedly in such good condition that much of the food it was carrying might still be intact in its storage jars. 'There are some broken jars around the wreck, but we believe that most of the amphorae inside the ship are still sealed and food-filled," Lt. Col. Francesco Schilardi of the police divers' group told the BBC of the containers. Local fisherman first became aware of the wreck when pieces of pottery began turning up in their

**Head 1**

An ancient Roman merchant vessel has been discovered off the Italian coastline, reportedly in such good condition that much of the food it was carrying might still be intact in its storage jars. 'There are some broken jars around the wreck, but we believe that most of the amphorae inside the ship are still sealed and food-filled," Lt. Col. Francesco Schilardi of the police divers' group told the BBC of the containers. Local fisherman first became aware of the wreck when pieces of pottery began turning up in their

**Head 2**

An ancient Roman merchant vessel has been discovered off the Italian coastline, reportedly in such good condition that much of the food it was carrying might still be intact in its storage jars. 'There are some broken jars around the wreck, but we believe that most of the amphorae inside the ship are still sealed and food-filled," Lt. Col. Francesco Schilardi of the police divers' group told the BBC of the containers. Local fisherman first became aware of the wreck when pieces of pottery began turning up in their

Figure 29: The next word is ***nets***. (A) Lyapunov sensitivity gives a high value for the word "in" which is a preposition and relevant to the next work "nets". Another relevant word "fisherman" is highlighted together with the object "pottery" in the nets. (B) Analysis of token relationships through attention matrices. We observe the same behaviors as the previous example.

