# OpenReview forum: "Neural ODE Transformers: Analyzing Internal Dynamics and Adaptive Fine-tuning"
_ICLR.cc/2025/Conference — ICLR 2025 Poster_

### Official Review · Reviewer_hGru · 2024-10-30

**Soundness:** 3
**Presentation:** 4
**Contribution:** 3
**Rating:** 8
**Confidence:** 4

**Summary:**

This work proposes an architecture that implements a Transformer using a Neural ODE approach (DiffEqFormer). The main contribution over existing Neural ODE works is that DiffEqFormer has time-dependent weights, as opposed to traditional weight-sharing. This equips DiffEqFormer with the interesting property of adaptive fine-tuning, i.e. baing able to fine-tune a model with N layers as a M layer version of itself. The authors show competitive performance compared to GPT-S/M/L on two datasets. Additionally, interpretability results are obtained using the Lyapunov exponent of the proposed dynamical system.

**Strengths:**

**Originality:**

* The time-dependent implementation proposed is both elegant and original in my opinion.
* Using the Lyapunov exponent for interpretability is an original idea, which is also very well adapted to the framework proposed.
* The idea of adaptive fine-tuning is also original to me, specially given how natural it is under the DiffEqFormer formulation.

**Quality:**

* The experimental setup related to interpretability of attention matrices and eigen-values is very complete.

**Clarity:**

* The paper is well written, with clear language. The mathematical notation and formulation is also clear and easy to follow. I want to congratulate the authors for that, I honestly was hesitant before reading the paper since many ODE papers are too heavy in notation. I really enjoyed reading this manuscript!

**Significance:**

* I believe this Neural ODE approach can have impact and can serve as starting point for further research. Both the idea of adaptive fine-tuning and time-dependent weights have strong research potential.

**Weaknesses:**

**Quality:**

* The experimental setup related to Language Modelling (6.1) uses arguably old models (GPT2). Although the experiments are useful to understand how DiffEqFormer is able to match performance, I missed a comparison with a current SOTA model. I believe this work could have more impact if a recent model was included, not needing an extremely large model, but maybe a 1B parameter LLM. For example,  [Llama-3.2-1b](https://huggingface.co/meta-llama/Llama-3.2-1B) or [Gemma-2-2b](https://huggingface.co/google/gemma-2-2b) could be excellent candidates.

**Clarity:**

* I felt that the discussion about _attractive_ and _repulsive_ scenarios (5.1) is a little bit disconnected from the rest, and much harder to parse. For example, I could not directly understand why a $V=-I_d$ behavior leads to model collapse. Also, in the conclusion of this section (L336) the authors emphasize that "_positive values from the QK pairs in Figure 2a and the copying behavior observed in the OV pair_" hint to the presence of induction heads in the last layer(s). However, I understood that what is important is the magnitude of QK pairs, and the positiveness of OV pairs. There is a mismatch between this conclusion and the claims in the explanation.  I encourage the authors to revisit this section and provide a clearer explanation of why the QK, and specially the OV analysis, is important.

**Significance:**

* As said, this work can be impactful given the applicability of DiffEqFormer. However, lacking a comparison with SOTA LLMs can diminish such impact.

**Questions:**

**Questions**:

* I encourage the authors to add a section (main body or appendix) about the computational differences betwee DiffEqFormer and a vanilla Transformer. For example, ML practitioners could be interested in training time, total number of parameters, FLOPs, GPU memory required and any other significant parameter the authors might think of, given their expertise in the field.

* In Section 5.2, I suggest including interpretability results using attention matrices, to make the differences claimed evident to the reader.

* In Section 6, can the authors elaborate on the choice of experimental setup? Why is GPT2 chosen among the breadth of language models available? Unless the constraints are compute-related, I believe this work could be much more impactful if DiffEqFormer was compared to a recent LLM architecture.
  * If some recent LLM was used, I am really intrigued in knowing about other metrics related to "_emergent capabilities_" are for DiffEqFormer. For example, does it perform better at 0-shot reasoning, on MMLU, etc? I am not asking the authors to provide all this analysis, but I definitely encourage them to include at least a comparison with a recent LLM.

* As far as I understand, the model is trained at fixed time intervals. I wonder whether the authors have thought about training at randomized time steps, as it is typically done for diffusion models.  Is it even feasible? Could this allow a more flexible adaptive fine-tuning later on? I would appreciate some discussion on this aspect during the rebuttal, for my better understanding of the work.

**Comments**:

* Overall, I believe this is a nice piece of research work. I enjoyed the reading and learnt throughout. My main concern is about the experimental setup. A comparison with some modern LLM would be very important to convince the community that DiffEqFormer is _equivalent_ to Transformer with the additional unique advantage of adaptability+interpretability. I am completely open to increase my review score upon a fruitful rebuttal.

* I would like to share my opinion about the experiments in Tables 1-2. Although the authors try to explain why GPT2-large is slightly better in terms of PPL for OpenWebText, I believe the strength of these results lie in DiffEqFormer being similar (not necessarily better) to GPT.

**Suggestions for readability + typos:**

* Very minor: Fig 1-b could have the arrow sin blue, as a connection with what is shown in Fig 1-a.
* L304: The OV matrix ~has~ have eigenvalues
* L322: model ~collapse~ collapses
* Fig 6 (legend) ~cratch~ scratch

---

> ### Author Response · Authors · 2024-11-24
>
> We sincerely thank the reviewer for their thorough review and encouraging comments. Below, we address your concerns comprehensively.
>
> ## Modern architecture
> Following the reviewer's suggestion, we have expanded our evaluation to include the Llama-1B architecture. In our comparative analysis, conducted with a training set of 20B tokens, our model demonstrates superior performance to Llama-1B, achieving a validation perplexity of **9.68** compared to Llama-1B's 10.17. Additionally, our model also shows stronger performance in downstream reading comprehension tasks.
>
> Our initial choice of GPT as a baseline was motivated by two points:
>
>  - Resource constraints when we start working on this idea of neural ODE for transformer
> - It would be more reasonable to validate DiffEqFormer's capabilities before scaling up
>
> We chose GPT specifically because it provides well-established benchmarks for smaller models, while newer architectures like Llama and Gemma primarily focus on large-scale implementations. However, we acknowledge that our current training setting (20B tokens) represents a relatively small scale compared to state-of-the-art Llama-1B implementations, which are trained on substantially larger datasets.
>
> ## Downstream task evaluation
>
> To demonstrate the emergent capabilities of LLMs, we provide the downstream evaluation (zero-shot and five-shot) in benchmarks including MMLU in Appendix C.7. Here are some notes on our model performance.
>
> In GPT-large Configuration, our model improves performance in summary as
> - Reading Comprehension:
>    - Lambada OpenAI (Zero-shot): +1.15% (37.24 → 38.39)
>    - Lambada Standard (Zero-shot): +1.07% (29.30 → 30.37)
>    - Lambada Standard (Five-shot): +1.77% (23.05 → 24.82)
> - Reasoning:
>   - Winogrande (Zero-shot): +1.82% (50.75 → 52.57)
>   - Winogrande (Five-shot): +4.34% (49.33 → 53.67)
>
>  - Scientific Knowledge:
>     - SciQ (Zero-shot): +1.50% (45.70 → 47.20)
>
> In Llama-1B Configuration, our model has a notable improvements in reading comprehension tasks:
>  - Lambada OpenAI (Zero-shot): +1.90% (41.84 → 43.74)
>  - Lambada OpenAI (Five-shot): +3.94% (30.47 → 34.41)
>  - Lambada Standard (Zero-shot): +3.97% (27.58 → 31.55)
>  - Lambada Standard (Five-shot): +4.18%s (24.04 → 28.22)
>
>
>
> ## Clarification for Section 5.1
> We thank the reviewer for raising the clarity issue in Section 5.1. We would like to clarify that when $V=-I$, particles in the system repel each other. Regarding our discussion of induction heads, we have revised the main text by providing a clearer description of their behavior, which consists of two key components: pattern matching (associated with QK pairs) and copying (associated with OV pairs). The simultaneous presence of positive eigenvalues in both matrices encourages the formation of induction heads according to  Elhage et al. (2021). We have updated the manuscript to reflect these clarifications.
>
> ## Computation aspects
> Regarding computation aspect, we provide additional ablation on GPU memory allocation as the same concern from other reviewers. We would like to note that our model may incur additional memory allocation, roughly less than 10% more GPU memory consumption.
>
> ## Randomized time steps
> We appreciate the suggestion regarding random time points for enhancing generalization during fine-tuning. We though about this approach but haven't looked carefully into it. However, we note potential challenges like adaptive step-size ODE solvers, which typically require careful step-size selection. Unlike diffusion models that use score-matching between arbitrary time points without ODE solutions, our model, or neural ODEs in general, may face stability issues with uneven step sizes. Nevertheless, we agree this direction is interesting for future investigation.
>
> ## Misc
> We included the attention-based explanation to compare with our Lyapunov exponent approach. We fixed the suggested typos.

---

> > ### Author Response · Authors · 2024-11-27
> > **Follow-up: Have our clarifications and revisions addressed the raised concerns?**
> >
> > Dear Reviewer hGru,
> >
> > Thank you for your valuable feedback. We have addressed your comments and made corresponding revisions to strengthen the manuscript. We welcome any additional suggestions and are happy to provide further clarification if needed.
> >
> > Please let us know if our revisions adequately address your concerns.
> >
> > Thank you for your time and consideration.

---

> > > ### Comment · Reviewer_hGru · 2024-11-27
> > > **Answer to Rebuttal**
> > >
> > > I want to thank the authors for a thorough rebuttal, which, in my opinion, has improved the quality of this work.  Notably:
> > >
> > > * Including Llama-1B as a representative of more modern LLMs, with good results.
> > > * Extended results on several downstream tasks, showing that the propsosed approach generalizes to different end tasks.
> > > * Ablation on GPU requirements.
> > > * Improved explanations.
> > >
> > > Given all the above, I recommend this paper for acceptance, raising my score to 8.

---

> > > > ### Author Response · Authors · 2024-11-28
> > > > **Thank you**
> > > >
> > > > Thank you for raising the score and your support for this paper. Your thoughtful feedback during the review process helped us strengthen the paper significantly. We appreciate your careful consideration and expertise in evaluating our work.

---

### Official Review · Reviewer_rn5p · 2024-11-02

**Soundness:** 3
**Presentation:** 2
**Contribution:** 2
**Rating:** 6
**Confidence:** 4

**Summary:**

This paper presents DIFFEQFORMER, an innovative framework that reformulates transformer architectures through the lens of non-autonomous neural ordinary differential equations (ODEs). The approach conceptualizes transformer layers as a continuous-time dynamical system, with the distinguishing feature of parameterizing attention and feed-forward weights as continuous functions of layer depth via neural networks.The authors conduct rigorous spectral analysis of the attention mechanisms, demonstrating increasing eigenvalue magnitudes across layers—a finding that challenges current theoretical frameworks predicated on weight-sharing assumptions. They further advance model interpretability by incorporating Lyapunov exponent analysis for examining token-level sensitivity.

**Strengths:**

1. The paper is exceptionally well-organized with a clear progression from theory to implementation. Complex mathematical concepts are presented in an accessible manner with effective visualizations.
2. The paper introduces an innovative approach using non-autonomous neural ODEs with time-dependent weights to model transformers, effectively overcoming limitations of previous weight-sharing methods. The comprehensive analysis through spectral dynamics of QK/OV pairs and investigation of clustering behavior provides deep theoretical insights into transformer dynamics while achieving competitive performance.
3. The introduction of Lyapunov exponents for analyzing token-level sensitivity is highly innovative. This provides new tools for model interpretability that go beyond traditional attention visualization methods.

**Weaknesses:**

1. The model's performance degradation at GPT-large scale raises concerns about its viability for larger architectures. The lack of experiments on billion-parameter scale models leaves uncertainty about whether this approach can effectively scale to sizes relevant for state-of-the-art language models.
2. While perplexity results are promising, the paper lacks zero-shot evaluation results which are crucial for understanding the model's practical capabilities. Additional metrics would better demonstrate the framework's advantages over traditional transformers.
3. The performance appears sensitive to the $d_{emb}$, but the paper provides limited guidance on how to optimally set this hyperparameter for different model scales.

**Questions:**

The time-dependent weight parameterization in your approach introduces additional parameters compared to standard transformer architectures. While you demonstrate comparable throughput, I'm concerned about the memory efficiency. Could you clarify the actual memory overhead during training and inference? Specifically, how does the memory usage scale with model size and demb.

---

> ### Author Response · Authors · 2024-11-24
>
> We sincerely thank the reviewer for their valuable feedback. Below, we address your concerns in detail.
>
> ## Large-Scale Model Performance
> Regarding large-scale models, we conducted additional investigations with recently acquired computational resources by scaling our model to 1B parameters using the Llama architecture. Our model demonstrates better performance compared to both the baseline Llama-1B and GPT-large. This improved performance stems from using more suitable optimization parameters—specifically, we found that our model benefits from a lower $\beta_2$ value in the Adam optimizer, which enables more flexible adaptation of time-dependent weights. These results confirm that our model shows comparable or better performance to GPT/Llama, though we acknowledge that further large-scale investigations would be valuable.
>
> ## Downstream task performance
>
> We conducted downstream task evaluations using `lm-evaluation-harness` in our updated manuscript. Please refer to Appendix C.7 for details.
>
> Our model shows similar performance across multiple benchmarks compared to baselines. It's worth noting the our model presents strong results in **reading comprehension tasks**. The following is a quick summary of improvements in our models.
>
> *GPT-large Configuration*:
>
> - Reading Comprehension
>
>   - Lambada OpenAI (Zero-shot): +1.15% (37.24 → 38.39)
>   - Lambada Standard (Zero-shot): +1.07% (29.30 → 30.37)
>   - Lambada Standard (Five-shot): +1.77% (23.05 → 24.82)
>
> - Reasoning
>
>   - Winogrande (Zero-shot): +1.82% (50.75 → 52.57)
>   - Winogrande (Five-shot): +4.34% (49.33 → 53.67)
>
>
> - Scientific Knowledge
>
>   - SciQ (Zero-shot): +1.50% (45.70 → 47.20)
>
> *Llama-1B Configuration*
>  - Reading Comprehension:
>
>    - Lambada OpenAI: +1.90% (Zero-shot: 41.84 → 43.74), +3.94% (Five-shot: 30.47 → 34.41)
>    - Lambada Standard: +3.97% (Zero-shot: 27.58 → 31.55), +4.18% (Five-shot: 24.04 → 28.22)
>
> These results highlight our model's particular strength in reading comprehension tasks across both architectures, with especially significant gains in the Llama-1B configuration.
>
> ## Computational Efficiency and Parameters
>
> While our model's total parameter count exceeds that of the GPT model, the effective parameters per batched input remain comparable to the baseline. This is because the time-dependent weight calculations $W(t)$ are batch-independent, computed before any batch-input operations. Regarding memory usage, our ablation study (Appendix D.1, Figure 16) demonstrates sub-linear memory scaling with respect to $d_{emb}$ dimension—a 150-dimension $d_{emb}$ increase results in less than 10% additional GPU memory usage. To address computational overhead, we believe promising directions include efficient representations such as sparse structure matrices (Monarch; Dao et al., 2022) and Block Tensor Train methods (Qiu et al., 2024).
>
> We acknowledge that the lack of clear guidelines for selecting optimal $d_{emb}$ values represents a limitation in our current work. Developing systematic approaches for determining these values presents an important direction for future research.
>
> ### References
>
> - Dao et al. (2022). Monarch: Expressive Structured Matrices for Efficient and Accurate Training
> - Qiu et al. (2024). Compute Better Spent: Replacing Dense Layers with Structured Matrices

---

> > ### Author Response · Authors · 2024-11-27
> > **Follow-up: Have our clarifications addressed the concerns?**
> >
> > Dear Reviewer rn5p,
> >
> > Thank you for your valuable feedback. We have addressed your comments and made corresponding revisions to strengthen the manuscript. We welcome any additional suggestions and are happy to provide further clarification if needed.
> >
> > Please let us know if our revisions adequately address your concerns.
> >
> > Thank you for your time and consideration.

---

> > > ### Author Response · Authors · 2024-12-01
> > >
> > > Dear reviewer rn5p,
> > >
> > > We wanted to follow up on our previous responses and check if they addressed your comments.
> > >
> > > If you have any questions or concerns, we would be happy to provide additional clarification.
> > >
> > > Best regards,
> > >
> > > Authors

---

> > > > ### Comment · Reviewer_rn5p · 2024-12-03
> > > >
> > > > I appreciate the authors' clarifications on zero-shot performance and large model comparison, which demonstrated their competitiveness. While I regret the lack of guidance on the $d_{emb}$ hyperparameter setting, I slightly raise my score to 6.

---

> > > > > ### Author Response · Authors · 2024-12-03
> > > > > **Thank you**
> > > > >
> > > > > Thank you for your response and for raising the score. We appreciate your time and feedback. We agree that developing a more principled approach to selecting $d_{emb}$ would strengthen the paper. We postulate that $d_{emb}$ has deep connections with the intrinsic dimensionality of both the data and model size. This represents an interesting yet challenging research direction for future work.

---

### Official Review · Reviewer_4Us1 · 2024-11-02

**Soundness:** 3
**Presentation:** 3
**Contribution:** 3
**Rating:** 6
**Confidence:** 4

**Summary:**

This paper presents a neural ODE-based approach for generating transformer weights. Unlike existing methods that use weight-sharing, this approach employs hyper-networks to produce the attention and feed-forward layer weights as functions of a continuous layer index (or time).

**Strengths:**

1) This paper employs flexible computation and dynamically adjusts the step size of its ODE solvers to fine-tune the model with variable layer lengths, achieving competitive performance with vanilla transformers. This adaptation is particularly promising as it allows for post-pretraining adjustments to the model's architecture, which is a great direction for future research.

2) They demonstrate the model's capacity through spectral analysis, revealing that increasing eigenvalue magnitudes inhibit cluster formation compared to weight sharing based prior works.

3) They also leverage the Lyapunov exponent to analyze token-level sensitivity for explainablity where they utilises both attention map and Jacobian which has similarity with the work in computer vision[1] for better faithfullness.

[1] Chefer, Hila, Shir Gur, and Lior Wolf. "Transformer interpretability beyond attention visualization." Proceedings of the IEEE/CVF conference on computer vision and pattern recognition. 2021.

**Weaknesses:**

1) No evaluation of downstream performance is conducted.

2) Training is computationally and memory-intensive; for instance, training a GPT-medium model requires 1.8B (with d_emb=144) parameters to achieve competitive performance, while the original GPT-2 medium model has only 345M parameters.

**Questions:**

Could the authors include evaluation on downstream tasks? This information would be valuable in assessing how well the model generalizes beyond the initial validation perplexity. They can show some zero and 5 shot evaluation on benchmarks from appendix of [2].

[2] Biderman, Stella, et al. "Pythia: A suite for analyzing large language models across training and scaling." International Conference on Machine Learning. PMLR, 2023.

---

> ### Author Response · Authors · 2024-11-24
>
> We sincerely appreciate the reviewer's thoughtful feedback. We have addressed your concerns as follows:
>
> ## Downstream task evaluation
>
> Following reviewer suggestions, we conducted comprehensive downstream task evaluations using `lm-evaluation-harness`. Detailed results are available in Appendix C.7.
>
> Our model demonstrates performance comparable to or better than baselines across multiple benchmarks, with particularly strong results in reading comprehension tasks. Here are the key findings:
>
> In GPT-large Configuration, our model improves performance in summary as
>
> *Reading Comprehension*:
> - Lambada OpenAI (Zero-shot): +1.15% (37.24 → 38.39)
> - Lambada Standard (Zero-shot): +1.07% (29.30 → 30.37)
> - Lambada Standard (Five-shot): +1.77% (23.05 → 24.82)
>
> *Reasoning*:
> - Winogrande (Zero-shot): +1.82% (50.75 → 52.57)
> - Winogrande (Five-shot): +4.34% (49.33 → 53.67)
>
> *Scientific Knowledge*:
> - SciQ (Zero-shot): +1.50% (45.70 → 47.20)
>
> In Llama-1B Configuration, our model has a notable improvements in reading comprehension tasks:
>
> - Lambada OpenAI (Zero-shot): +1.90 points (41.84 → 43.74)
> - Lambada OpenAI (Five-shot): +3.94 points (30.47 → 34.41)
> - Lambada Standard (Zero-shot): +3.97 points (27.58 → 31.55)
> - Lambada Standard (Five-shot): +4.18 points (24.04 → 28.22)
>
> These results demonstrate our model's particular strength in **reading comprehension tasks** across both architectures, with substantial improvements in the Llama-1B configuration.
>
> ## Computational Resources and Parameter Counts
> We agree that our model has a higher total parameter count than the GPT model. However, the number of effective parameters per batched input remains roughly equivalent to the baseline model. This is because the parameters from time-dependent weight calculations $W(t)$ are batch-independent, as these weights are computed prior to any batch-input operations. Regarding memory usage, although we require additional storage for time-dependent weight parameters, our ablation study (Appendix D.1, Figure 16) demonstrates sub-linear memory scaling with respect to $d_{emb}$ dimension—increasing $d_{emb}$ to 150 results in only a <10% increase in GPU memory usage. We would think that approaches using more efficient representations, such as sparse structure matrices (Monarch; Dao et al., 2022) and Block Tensor Train methods (Qiu et al., 2024) can be good directions to address these issues.
>
> ### Reference:
>
> - Dao et al 2022, Monarch: Expressive Structured Matrices for Efficient and Accurate Training
> - Qiu et al, 2024 Compute Better Spent: Replacing Dense Layers with Structured Matrices

---

> > ### Author Response · Authors · 2024-11-27
> > **Follow-up: Have our clarifications addressed the concerns?**
> >
> > Dear Reviewer 4Us1,
> >
> > Thank you for your valuable feedback. We have addressed your comments and made corresponding revisions to strengthen the manuscript. We welcome any additional suggestions and are happy to provide further clarification if needed.
> >
> > Please let us know if our revisions adequately address your concerns.
> >
> > Thank you for your time and consideration.

---

> ### Author Response · Authors · 2024-12-01
>
> Dear Reviewer 4Us1,
>
> As we are approaching the end of the discussion period, we wanted to follow up on our previous responses and ensure they sufficiently addressed your comments.
>
> If you have any remaining questions or concerns, we would be glad to provide additional clarification.
>
> Best regards,
>
> Authors

---

### Official Review · Reviewer_Pfg2 · 2024-11-04

**Soundness:** 2
**Presentation:** 2
**Contribution:** 2
**Rating:** 3
**Confidence:** 3

**Summary:**

This paper proposes a novel approach to modeling transformer architectures using depth-varying neural ordinary differential equations (ODEs). The authors introduce time-dependent weights for attention and feed-forward blocks, implemented through neural networks that take a continuous layer index ($t$) as input. They analyze the model's dynamics through spectral analysis and Lyapunov exponents, aiming to provide insights into transformer behavior. The work also explores adaptive fine-tuning capabilities enabled by the ODE formulation, showing comparable performance to GPT models across various configurations and datasets.

**Strengths:**

- The paper makes an interesting attempt to bridge the gap between theoretical analysis of transformers and practical implementations by using neural ODEs with depth-varying weights.
- The proposed adaptive fine-tuning approach, which leverages the discussed ODE formulation to enable flexible architecture changes during fine-tuning, presents a potentially useful direction for model adaptation.
- The presentation includes detailed visualizations of spectral dynamics and Lyapunov exponent analysis, which helps readers understand the model's behavior.

**Weaknesses:**

1. Limited Technical Novelty:
- The core formulation of transformers as ODEs is directly borrowed from Lu et al. (2019) without significant theoretical advancement.
- The analysis methodology for the dynamics closely follows Geshkovski et al. (2023a,b) without introducing substantial new analytical tools or insights.
- The time-dependent weight parametrization, while interesting, lacks theoretical justification for its specific design choices.
2. Insufficient Analysis Rigor:
- The spectral analysis lacks formal theoretical guarantees or concrete connections to model performance.
- The Lyapunov exponent analysis is essentially equivalent to analyzing the Jacobian of transformer blocks, which varies with each input and doesn't provide generalizable insights.
- Many claims about model behavior are supported by cherry-picked empirical observations without theoretical justification.
3. Experimental Limitations:
- The performance improvements over baseline models are modest and inconsistent across different settings.
- The ablation studies don't sufficiently justify the architectural choices, particularly the dimension of time embeddings.
- The paper oscillates between being an analysis paper and an empirical paper without excelling in either direction.

4. Presentation Issues:
- The paper structure is somewhat disjointed, with early sections (1-4) largely reiterating existing work.
- Many design choices appear arbitrary and lack proper motivation or ablation studies.
- The relationship between the theoretical analysis and practical benefits isn't clearly established.

**Questions:**

1. Could you provide theoretical justification for the specific design of the time-dependent weight parametrization? What motivated these particular choices, and how do they relate to the model's performance?
2. The paper presents both analytical tools (spectral analysis, Lyapunov exponents) and empirical results, but the connection between them isn't clear. How do the insights from your analysis inform or explain the observed performance in the experiments?
3. The adaptive fine-tuning capability seems promising, but its benefits aren't fully explored. Could you elaborate on scenarios where this flexibility provides concrete advantages over traditional fine-tuning approaches?
4. Given that the Lyapunov exponent analysis is input-dependent, how generalizable are the insights gained from this analysis? What patterns or principles have you observed across different classes inputs that might provide more universal understanding of transformer behavior?

---

> ### Author Response · Authors · 2024-11-24
>
> We thank the reviewer for their time and thoughtful feedback. Below we address each concern:
>
> ## Relationship to Lu et al. 2019 and Dynamic Particle Systems
> We acknowledge that viewing transformers as dynamic particle systems originated with Lu et al. 2019, a perspective that has influenced both theoretical and practical approaches (e.g., Dutta et al. 2021, Geshkovski et al. 2023a,b). As clarified in Section 3, our work differs from Lu et al.'s MacaronNet in two key aspects. First, while MacaronNet was inspired by ODE solvers to create a transformer architecture with two feed-forward blocks and one attention layer, it **remains a discrete-layer neural network**. This limits its adaptive fine-tuning capabilities. In contrast, our model is a true neural ODE, functioning as a **continuous-layer neural network**. This allows us to leverage neural ODE advantages, particularly in adaptive fine-tuning.
>
> ## Distinction from Geshkovski et al. (2023)
> We emphasize that our analysis provides compelling evidence that clustering behavior may not occur in autoregressive transformer models. This conclusion is derived from studying a realistic model that achieves performance comparable to GPT.
> Our work complements the theoretical analyses presented in Geshkovski et al. (2023a,b) by offering a practical, empirically-grounded approach. The key distinction is that our analysis stems from studying DiffEqFormer—a model that demonstrates real-world effectiveness by matching GPT's performance. The insights gained from our model suggest new directions for theoretical dynamical systems research and can help refine the assumptions used in studies like Geshkovski et al. (2023a,b) to better align with practical implementations.
>
> ## Model Performance Updates
> At the time of submission, our model showed slightly lower performance than GPT-large. We identified that this was due to using identical optimizer settings across different architectures without proper hyperparameter tuning. After conducting further optimization, we adjusted the Adam optimizer by decreasing the $\beta_2$ parameter. This modification led to improved performance:
>
> *GPT-large configuration*:
> - Our model's validation perplexity: **15.43**
> - GPT baseline perplexity: 15.64
>
> *Llama-1B configuration*:
> - Our model's validation perplexity: **9.68**
> - Llama baseline perplexity: 10.17
>
> We have updated these results in the paper and conducted additional evaluations on downstream tasks as suggested by reviewers. The new results demonstrate performance gains over the baseline mode in reading comprehension tasks. For detailed results and analysis, please refer to Appendix C.5 and Appendix C.7.
>
> ## Time-Dependent Weight Design
> Our time-dependent weight construction comprises three key components. First, we employ sinusoidal embedding, utilizing Fourier features that align with our continuous-time layer indexing. Second, we incorporate an MLP block to ensure non-linearity. This is a design inspired by score functions in diffusion models. Our motivation stems from the established connection between ODEs and diffusion models. The third and distinguishing component of our approach is the weight projection, which transforms embeddings into constructed weights.
>
> ## Value of Spectral Analysis and Lyapunov Exponents
> While our spectral analysis and Lyapunov exponent studies do not directly translate to model performance improvements, they offer valuable interpretability insights. Our model not only achieves performance comparable to GPT but also provides enhanced interpretability—a crucial advantage given the challenges of understanding large language models. The key benefit lies in our continuous weight design: analyzing the time-dependent weight function $W(t)$ offers a more attractive approach to understanding model properties compared to examining multiple discrete layers in GPT. Our analysis draws from classical dynamical systems theory, providing a novel framework for understanding transformer architectures. This continuous perspective offers a more elegant and mathematically grounded approach to model analysis.

---

> ### Author Response · Authors · 2024-11-24
>
> ## Practical Applications of Adaptive Fine-tuning
> Adaptive finetuning can offer a practical solution for deploying models under hardware constraints. Consider this scenario:
> - Initial Training:
>     - Model is pretrained with 18 time steps (layers)
>     - Training performed on high-performance servers
> - Deployment Adaptation:
>   - Target environment: Less powerful inference servers
>   - Adaptively finetune to reduce to 12 layers
>   - For example, (18→12) configuration can describe this scenario
>
> While this process shares similarities with knowledge distillation, our adaptive finetuning approach is distinct. The reduced-resource model differs from the pretrained version only by the numerical error introduced by the solver's step size adjustment.
>
> Additionally, time-dependent weights can serve as an effective initialization strategy for large-scale discrete-layer transformers. This approach is particularly valuable when starting from a pretrained time-dependent model trained with limited computational resources. The weights can be transferred and adapted to initialize larger models, providing a resource-efficient pathway for scaling up transformer architectures.
>
> ## Lyapunov Exponent Analysis
> Lyapunov analysis provides a more generalized and principled approach compared to traditional perturbation methods. To calculate the Lyapunov exponent, we must solve an ordinary differential equation (ODE) where the vector field is determined by the Jacobian of the output tokens with respect to the input tokens. This Jacobian matrix is influenced by two key factors:
> - The attention weight distribution across input tokens
>  - The embedding distance between input and output tokens (detailed in Appendix B.2)
>
> This dual dependence demonstrates that the Lyapunov exponent captures more complex dynamics than what can be inferred from attention patterns alone (Jain and Wallace).

---

> > ### Author Response · Authors · 2024-11-27
> > **Follow-up: Have our clarifications addressed the raised concerns?**
> >
> > Dear Reviewer Pfg2,
> >
> > Thank you for your valuable feedback. We have addressed your comments and made corresponding revisions to strengthen the manuscript. We welcome any additional suggestions and are happy to provide further clarification if needed.
> >
> > Please let us know if our revisions adequately address your concerns.
> >
> > Thank you for your time and consideration.

---

> ### Author Response · Authors · 2024-12-01
>
> Dear Reviewer Pfg2,
>
> As the discussion period is coming to an end, we wanted to check again if our responses addressed your concerns.
>
> We greatly value your feedback and would be happy to provide any clarifications.
>
>
>
> Best regards,
>
> Authors

---

### Author Response · Authors · 2024-11-24
**General Comment**

Thank you for your thoughtful feedback on our paper. We appreciate the time you've taken to review our work and have addressed your general concerns comprehensively below.


## Downstream Evaluation Results
We acknowledge the reviewers' shared concern regarding thorough baseline comparisons in downstream evaluations. We have now conducted both zero-shot and five-shot evaluations to address this:


| Benchmark | GPT (Zero-shot) | Our Model (Zero-shot) | GPT (Five-shot) | Our Model (Five-shot) |
|-----------|----------------|-------------------|----------------|-------------------|
| ARC-Challenge | 20.65 (1.18) | 20.90 (1.19) | 21.84 (1.21) | 21.25 (1.20) |
| ARC-Easy | 36.03 (0.99) | 35.82 (0.98) | 37.12 (0.99) | 36.32 (0.99) |
| HellaSwag | 25.53 (0.44) | 25.61 (0.44) | 25.40 (0.43) | 25.52 (0.44) |
| Lambada OpenAI | 37.24 (0.67) | **38.39** (0.68) | 27.79 (0.62) | 27.83 (0.62) |
| Lambada Standard | 29.30 (0.63) | **30.37** (0.64) | 23.05 (0.59) | **24.82** (0.60) |
| LogiQA | 21.35 (1.61) | 21.35 (1.61) | 20.74 (1.59) | 21.81 (1.62) |
| MMLU | 22.95 (0.35) | 22.99 (0.35) | 26.08 (0.37) | 25.10 (0.36) |
| PIQA | 56.26 (1.16) | 56.96 (1.16) | 55.82 (1.16) | 56.69 (1.16) |
| SciQ | 45.70 (1.58) | 47.20 (1.58) | 50.30 (1.58) | 50.90 (1.58) |
| Winogrande | 50.75 (1.41) | **52.57** (1.40) | 49.33 (1.41) | **53.67** (1.40) |
| WSC273 | 55.31 (3.01) | 53.11 (3.03) | 50.92 (3.03) | 53.11 (3.03) |

*Performance comparison between GPT and our model in zero-shot and five-shot settings.*


| Benchmark | Llama-1B (Zero-shot) | Our Model (Zero-shot) | Llama-1B (Five-shot) | Our Model (Five-shot) |
|-----------|-------------------|-------------------|-------------------|-------------------|
| ARC-Challenge | 22.27 (1.22) | 22.27 (1.22) | 21.76 (1.21) | 22.44 (1.22) |
| ARC-Easy | 29.67 (0.94) | 31.10 (0.95) | 30.01 (0.94) | 31.44 (0.95) |
| HellaSwag | 25.20 (0.43) | 25.32 (0.43) | 25.19 (0.43) | 25.36 (0.43) |
| Lambada OpenAI | 41.84 (0.69) | **43.74** (0.69) | 30.47 (0.64) | **34.41** (0.66) |
| Lambada Standard | 27.58 (0.62) | **31.55** (0.65) | 24.04 (0.60) | **28.22** (0.63) |
| LogiQA | 21.51 (1.61) | 22.27 (1.63) | 21.04 (1.60) | 21.20 (1.60) |
| MMLU | 22.93 (0.35) | 22.98 (0.35) | 26.43 (0.37) | 24.72 (0.36) |
| PIQA | 53.26 (1.16) | 53.81 (1.16) | 53.65 (1.16) | 53.10 (1.16) |
| SciQ | 31.70 (1.47) | 33.20 (1.49) | 34.20 (1.50) | 34.80 (1.51) |
| Winogrande | 50.51 (1.41) | 49.64 (1.41) | 51.38 (1.40) | 49.33 (1.41) |
| WSC273 | 53.11 (3.03) | 52.38 (3.03) | 52.01 (3.03) | 50.92 (3.03) |

*Performance comparison between *Llama-1B* (new configuration) and our model in zero-shot and five-shot settings.*



We found that our model outperformed baselines (GPT-large, Llama-1B) in **reading comprehension tasks** (Lambada OpenAI and Lambada Standard). In other tasks, our model is comparable to baselines. For further details, please refer to Appendix C.7.

## Performance Optimization Improvements
Regarding the noted performance inconsistencies, particularly the slight underperformance in the GPT-large setting, we have identified the difference of the loss landscape between our model and the baselines as the root cause. We use a new configuration of Adam optimizer as:

- Previous Configuration: Adam optimizer with $\beta_1=0.9, \beta_2=0.999$
- Updated Configuration: Adam optimizer with $\beta_1=0.9, \beta_2=0.95$

The updated configuration leads to better performance (from 15.89 to **15.43**) compared to baseline across both optimizer settings.
We attribute this to the fact that the time-dependent weight $W(t)$’s role in generating all weights for all layers, which *benefits from more flexible parameter adaptation*. The reduced $\beta_2$ value enables faster forgetting of gradient histories, enhancing this adaptation. For a detailed analysis, please refer to Section C.5 in our revision.

## Llama-1B Architecture Results
As suggested by **Review hGru**, we  evaluate our model's scalability and architectural flexibility by conducting additional experiments with the Llama-1B architecture. Our model has a significant performance improvements over the Llama-1B baseline (**9.68** vs 10.17). Both are trained on 20B tokens of Open Web Text dataset. Our model also enhanced performance in reading comprehension tasks like GPT-large configuration.

## Revisions Overview
Our revision includes:
- Comprehensive downstream task analysis (detailed in Experiments section with supporting figures and tables in Appendix)
- Complete Llama-1B experimental results (Appendix C.6)
- Detailed GPU memory allocation analysis
- Attention-based explanation in comparison with Lyapunov exponent
- Source code updated according to new updates.

---

### Author Response · Authors · 2024-12-04
**Post-discussion message**

Dear AC and Reviewers,

We sincerely appreciate your comprehensive review of our paper and would like to provide this summary as the discussion period concludes.

Our work advances the key connection between neural ODE and Transformer research fields through the following contributions:

- Demonstrate that our proposed _continuous-layer_ transformers can achieve performance that **matches or outperforms** equivalent _discrete-layer_ transformers.
- Introduce an **adaptive fine-tuning approach** for dynamic layer depth adjustment in fine-tuned models.
- Enhance **model interpretability**  leveraged by  the continuous-time formulation of weights. It enables sophisticated analysis like spectral analysis and Lyapunov exponents, providing insights that can be challenging to obtain with traditional discrete transformers.

In response to the reviewers' feedback, we have thoroughly addressed the following key areas of concern.
- We expanded our evaluation to include downstream tasks as suggested by Reviewers **4Us1**, **m5p**, and **hGru**
- We validated our model's consistent performance across various settings, responding to concerns from Reviewers **4Us1** and **Pfg2**.
- Following guidance from Reviewers **m5p** and **hGru**, we scaled up the model size and investigated alternative transformer architectures.

We believe these revisions strengthen our contribution to the field.

Thank you for your valuable time and consideration throughout this process.

Best regards,

Authors

---

### Meta-Review · Area_Chair_sUKB · 2024-12-22

**Metareview:**

## Summary
This paper introduces a transformer model based on neural ordinary differential equations (ODEs). It uses neural networks to optimize attention and feed-forward weights as continuous functions of layer depth. The authors analyze the model’s dynamics using spectral analysis, observing increasing eigenvalue magnitudes, which question weight-sharing assumptions in theoretical studies. They also use the Lyapunov exponent to assess token-level sensitivity, enhancing interpretability. The proposed model matches GPT’s performance across various settings and offers adaptable fine-tuning capabilities.

## Decision
The paper is very well-written and has a very clear presentation. It introduces a novel approach using non-autonomous neural ODEs with time-dependent weights to model transformers, which overcome the shortcomings of previous weight-sharing methods. The comprehensive analysis and theoretical insights into transformer dynamics while achieving competitive performance are very valuable contributions of this paper. The use of Lyapunov exponents for analyzing token-level sensitivity is also novel. The reviewers raised some important concerns during the rebuttal period, but the authors have done a good job addressing them. Thus, I recommend acceptance of this paper. The reviewers have raised some important concerns during the rebuttal period, I recommend the authors to address all those concerns for the camera-ready upon acceptance.

**Additional Comments On Reviewer Discussion:**

The reviewers have provided some crucial concerns related to experiments, evaluations, and work related to this paper. The authors did an excellent job addressing those issues. Reviewer Pfg2 has raised some important concerns, suggesting the rejection of the paper, but I think the authors have done a good job of addressing those concerns. The reviewer was not responsive during the rebuttal phase, despite asking for their input to respond to the rebuttal. Thus, I recommend this paper for acceptance.

---

### Decision · Program_Chairs · 2025-01-22

Accept (Poster)